

# Statistical Analysis of Wave Energy Resources Available for Conversion at Natural Caves of Cape-Verde Islands.

**W. M.L. Monteiro[1] A. J. Sarmento[2], A.J. Fernandes[1] and J.M. Fernandes[1]**

[1]{Department of Science and Technology, University of Cape-Verde, Praia, Cape-Verde}

[2]{Lisbon High Technical Institute, Technical University of Lisbon, Lisbon, Portugal}

Correspondence to: W. M.L. Monteiro (wilson.monteiro@docente.unicv.edu.cv)

**Abstract**

Using the time-series of significant wave height and the peak period between 1979 and 2009 generated by SOWFIA project, some relevant statistical information about energy content available in ocean waves in Cape-Verde is obtained. The monthly and annual time-series of the average power are analysed and the confidence intervals for their values are defined. Considering all of the 31 years of data, the results show that the most energetic month, from the average power point of view is January (23.49 kW/m) and the least energetic month is July (15.04 kW/m). In fact, the monthly average power decays from January to July and increases from July to December (21.21 kW/m). The annual average power exhibits a clear attenuation over the 31 years analysed, the reason for which is not yet clear to us. However, using the appropriate Autoregressive Integrated Moving Average (ARIMA) model it is possible to estimate that future values of the annual average power tend to oscillate around 18.2 kW/m. Through the Coefficient of Variation of Power (COVP), obtained by dividing the standard deviation of the power time-series by the average power, it is possible to conclude that the wave resource is stable, with COVP between 0.46 and 0.66. The values of the Monthly Variation Index (MVI), the maximum range of the monthly mean wave power relative to the yearly mean level, show that the resource is relatively stable, with MVI < 1.2. The present work calculates the available power input into the Natural Caves (NCs) in Cape Verde Islands, through a rigorous analysis of the wave climate that excites them. The minimum sampling size and the corresponding numbers of days of measurements per month,



are also estimated.

## 2   **1   Introduction**

Ocean waves constitute one of the renewable sources of energy that are gradually entering the
market of clean and sustainable energy worldwide. The global theoretical energy from ocean
wave is estimated in $8 \times 10^6$ Twh/year (Boyle, 2004). Many countries around world have been
investing on this natural resource to produce useful and sustainable energy. Portugal (Pelamis
and Pico Plant Projects.), Australia (CETO and OCEANLIX projects), France (SEAREV
project), UK (OYSTER WEC and Limpet projects) and Holland (AWS project) are examples
of some countries that have recognized the feasibility of harvesting this source of energy
(ABP, 2004). According to the International Renewable Energy Agency (Monford et al.,
2014), around 64 % of the Wave Energy Converters (WECs) has been projected for offshore
application and 36% for near-shore and onshore operation. Some full-scale operational tests
have been realized. These include the OYSTER device from Aquamarine Power, the Wave
Roller from AW-Energy, Pelamis P2 from the Pelamis Wave Power, the Seabased and the
Sea-Tricity devices. Magagna (2011) has identified, in 2011, over 100 wave energy
developers. Yet, EMEC (2014) has listed 170 wave energy developers worldwide. About 45%
of the wave energy developers are based in or are currently developing projects in the
European Union (EU) regions. The global installed capacity of wave energy remains low and
the technologies are still at an advanced R&D stage. Just a few machines have sustained long
operational hours, such as the Aquamarine OYSTER (>20000 hours) and Pelamis (cumulative
> 10000 hours) (Scottish Renewable, 2014). The growth of the wave energy sector is lower
than expected and this situation may affect the confidence of investors in this area. Success in
attracting future Original Equipment Manufacturer investments will depend on the capacity of
the developers in improving performance, reducing cost and validating wave energy
technologies. The long-term global wave energy is expected to become cost competitive and
provide an alternative to other Renewable Energy Sources and conventional energy resources.
Through a review of the existing data available, the different cost components in the Capital
Expenditure (CAPEX) estimate for wave energy have been identified as follows (Table 1):




Table1. Costs components estimate for wave energy extraction (JRC, 2014).

| | |
|---|---|
| Civil and Structural costs | 38% |
| Major Equipment costs | 42% |
| Electrical and I&C supply and installation costs | 8% |
| Project indirect cost | 7% |
| Development cost | 5% |

Thus, the main contributors to the CAPEX are mechanical equipment, civil and structural
costs. In this context, the developers of wave energy technologies must undertake efforts and
strategies aimed at reducing mainly the two above mentioned costs.
SOWFIA-Streamlining of Ocean Wave Farm Impact Assessment is an EU Intelligent Energy
European Project with the goal of sharing and consolidating pan-European experience and
best practices for consenting processes and environmental and socio-economic impact
assessment (IA) for offshore wave energy conversion developments. This project brings
together ten partners across eight EU Member States actively involved in planned wave farm
test centers and aims at providing recommendations for streamlining of IA approval processes
with the purpose of removing legal, environmental and socio-economic barriers associated
with development of the wave energy farms.
Cape-Verde is an archipelago of ten islands in the Atlantic Ocean, off the West Coast of
Africa, with roughly half million people. The country is totally dependent on oil to produce
electricity, having one of the most expensive cost of electricity in Africa, around 0.28
Euro/kWh (Electra, 2012) versus 0.17 Euro/kWh (Senelec, 2015) at Senegal, a continental
neighbour. Some investments were made by the Government with the purpose of introducing
renewable sources of energy in the country, basically solar and wind energy. The Government
has defined an ambitious goal that consists in achieving 50% of Renewable Energy
penetration in the country by 2020 (GESTO, 2011). Some research on using ocean energy
through the OTEC – Ocean Thermal Energy Conversion system and WaveStar technology
(Wave energy) was initiated in the country but these projects still lack feasibility studies.



However, four projects for offshore wave energy conversion based on the Pelamis technology
were proposed for four of the islands (GESTO, 2011): Sal (3.7 MW), S. Antão (3.7 MW),
S.Vicente (3.7 MW) and Boavista (3.5 MW).
Being composed of islands, most of Cape-Verde's economic activities (around 90%) are
concentrated on coastal areas (Carvalho, 2013). In this context, it makes sense to use wave
energy for producing electricity locally. A clear alternative is harvesting Natural Caves
existing just below the rocky shore, with fountain-like structures (Fig. 1).
NCs are caverns that form naturally under the rocky shorelines, inside of which there is an air
layer. This air layer acts like an air pump as the wave enters and leaves these natural
infrastructures. As a result, the air is forced to go in and out of the NCs, through surface holes
that exist on top of the cave. Fig. 2 shows a Natural Caves with two holes in operation.
Monteiro and Sarmento (2015) carried out a study aiming at characterizing the NCs in the
context of wave energy extraction.

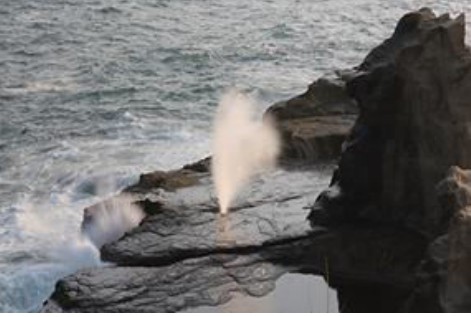
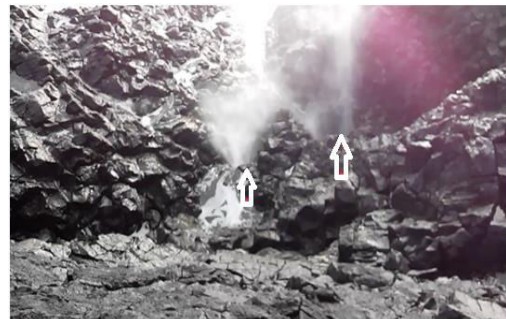

Figure 1. Activity in a Natural Cave.  Figure 2. Activity in a NCs with two holes.

The Principles of NCs operations are similar to the man-made Oscillating Water Column
device, projected for onshore application.
The justification for the idea of using the NCs is the possible cost reduction on the Civil and
Structural components, which are, as mentioned before, one of the most important costs
associated with building wave energy devices to produce electricity, and also to minimize the



risks of collapse, by taking advantage of the sturdiness of the natural rocky structure, time
tested by the waves and storms.
To evaluate the potential of NCs for electricity production, it is necessary to estimate its
output power. To do this, a set of experiments aimed at determining the values of some
important physics parameters of NCs operations need to be conducted. Monteiro and
Sarmento (2015) carried out the analytical modelling of the NCs operations as a function of
their functioning physical parameters. The present study is part of a deeper work aimed at
quantifying the output power of NCs and to project an adequate power take-off system to be
adapted on their holes, for electricity production.
Since the excitation waves are irregular, non-linear and non-stationary phenomenon it is very
important to determine beforehand the sampling size, i.e. how long it takes to carry out the
experiments on NCs to guarantee the time representativeness of its output power. To achieve
this goal, some statistical analysis has to be carried on the wave energy input regime.

## 15  2   Methodology

Calculation of the wave energy input regime is carried out using principles and parameters
described below.

### 18  2.1   Average Power

In deep water, where the depth is greater than a half of the wavelength, the average wave
power can be determined through the following equation, applied only for unidirectional
Pierson-Moskowitz wave spectrum.

$$P = \frac{\rho g^2 H_s^2}{64\pi} T_e \tag{1}$$

Where, $H_s$ is significant wave height, $T_e$ is energy period, defined in terms of the spectral
momentum by the following relation:



$$T_e = \frac{m_{-1}}{m_0} = \frac{\int\limits_{0}^{2\pi}\int\limits_{0}^{\infty} f^{-1}S(f)dfd\theta}{\int\limits_{0}^{2\pi}\int\limits_{0}^{\infty} S(f)dfd\theta}$$
(2)

in which, $m_{-1}$ is the spectral momentum of order -1, $m_0$ is the spectral momentum of order 0,
$f$ is the frequency, $S(f)$ is the spectral density function and $\theta$ is the direction of the
energy propagation (Dean and Dalrymple, 1991).
The characterization of the wave climate is made by the combination of the significant wave
height $H_S$ and peak period $T_P$ or the zero-crossing period $T_Z$ parameters. The energy period
determined by the Eq.(2) require the knowledge of the form of energy spectrum. When the
form of the energy spectrum is unknown it can be approximated by the some model as for
example, the Pierson-Moskowitz spectrum. This is the approximation used on the elaboration
of the Marine Atlas of Renewable Resources in UK (ABP, 2004). Another approximation
commonly used for $T_e$ is represented by $T_e \approx \alpha T_P$, where $\alpha$ is an empirical parameter. For
Pierson-Moskowitz spectrum, $\alpha = 0.86$ (Dean and Dalrymple, 1991). To evaluate the wave
resource for South of New England, Hagernam (2001) used the approximation $T_e = T_P$ and
considered this approximation very appropriate to make a preliminary analysis of wave
energy resource.
Using the monthly series of the available power in waves it is possible to define the annual
time-series of this parameter through the following expressions:

$$P_{aj} = \frac{\sum\limits_{i=initial\ month}^{initial\ month\ +11} P_{ij}}{12}$$
(3)

In the above equation $P_{aj}$ is the average power for year $j$, $P_{ij}$ is the average power for the
month $i$ and year $j$. In this way, the monthly time-series begin on January and ends on
December of each year.
It is important to note that there is no physical justification for wave power to be monthly
periodic, but since the sun-cycle is the underlying cause for atmospheric pressure distribution
and wind patterns over the ocean, most likely it will be yearly periodic.



The reason to calculate monthly series of available power is just related to how data is
collected and made available at SOWFIA.
## 2.2  Monthly Variation Index (MVI)
The temporal variability of the wave resources is a key factor that affects decisively the
feasibility of wave energy projects. In this sense, the regions of the ocean where the resources
are stable are more attractive for all possible investors. Naturally, the level of the average
power is another important factor for viability of wave energy harvesting.  The Monthly
Variation Index is defined as the ratio of the differences between the maximum and minimum
values of the monthly average wave power in year j by the corresponding annual average
wave power (Cornett, 2008). That is.
$$MVI_j = \frac{\left(P_{max} - P_{min}\right)_j}{P_{aj}}$$  (4)
where $P_{max}$ and $P_{min}$ are, respectively, the maximum and minimum values of the monthly
average power in year $j$.
## 2.3  Coefficient of Variation of Power (COVP)
COVP is another very important parameter used to evaluate the temporal variability of wave
resources. This quantity is defined by the ratio between the standard deviation of the wave
power and the respective annual average wave power in year j (Cornett, 2008).
$$\left(COVP\right)_j = \frac{\sigma\left(P(t)\right)_j}{P_{aj}}$$  (5)
In the Eq.(5), $\sigma\left(P(t)\right)_j$ represent the standard deviation of temporal series of wave power,
$P(t)$, for year $j$, and  $P_{aj}$ is the respective annual average wave power. According to Cornett
(2008), small values of COVP means that the wave resources are stable. For
$0.8 \le COVP \le 0.9$ the wave resources can be considered moderately instable. Therefore, for
$COVP > 0.9$ the resource is unstable.



## 2.4   Statistical analysis

The wave climate at a certain location is well characterized by the time-series of significant wave height and the peak period. Through these parameters that are recorded for each 3 hour (time interval necessary for verifying significant change in wave spectrum) other parameters such as the time-series of the average available power in waves can be defined. To understand the time-series behavior of some important wave parameters, to calculate the confidence interval, the smoothing curves, and the forecast of its values, many statistics tools of analysis are used. In this context, some well known statistics software such as XLSTAT and Minitab are used. Aspects such as the trend analysis, stationarity and normality tests of the average power are here analysed. To perform the forecast of the average power in waves, the Autoregressive Integrated Moving Average (ARIMA) model is used. Non-seasonal ARIMA model is generally represented as ARIMA (p,d,q) where, p is the order of the Autoregressive Model, d is the degree of differencing and q is the order of the Moving Average Model (Bisgaard and Kulahci, 2011).

Finally, the wave histogram is a table that lists the occurrence of the sea-states in terms of significant wave height and peak period or mean up-crossing period. It is the long-term statistical representation of sea states. Using the information in the wave histogram it is possible to identify the most common sea states in a certain region.

## 2.5   Representativeness of the monthly average output power from the NCs

The energy that excites the NCs is a function of the local wave regime, while its output energy depends on the input energy (wave regime) and on the geometry of the NCs (Fig. 3). For each NC the geometry is fixed, hence the output energy is directly influenced by the local wave regime. This mean that the variation in the output energy content is just caused by the variation in the input energy content, that is by the variation of the local wave regime. In this context, it is reasonable to assume that the minimum sampling size necessary for characterizing the input energy content is equal to the minimum sampling size needed to characterize the output energy from the NCs. The calculation of the minimum sampling size for characterizing the input energy into the cave is done using the Minitab Software. For three hours time interval between successive readings, the total number of data points acquired during one day is eight. So, if this minimum sampling size is represented by $N_{in}$, the



correspondent minimum time duration for data acquisition to achieve the representativeness
of the input power is $N_{in}/8$ days. Therefore, to guarantee the representativeness of the output
energy from the NCs the duration necessary to realize the experimental study on these natural
infrastructure is equal to $N_{in}/8$ days.

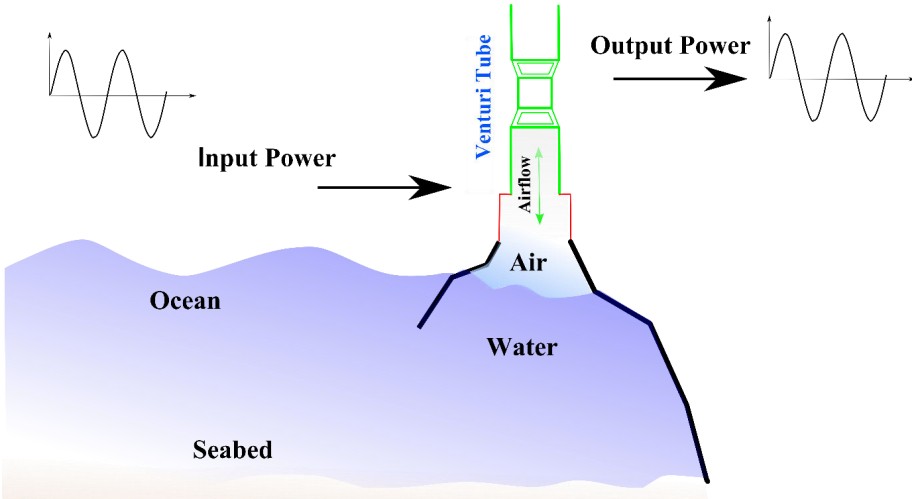

Figure 3. Energy production system by NCs.
**3    Results**
The information about the significant wave height (Hs) and peak period (Tp) for the wave
regime in Cape-Verde is obtained for the location characterized by the coordinate 16ºN-24ºW,
where the water depth is around 3.7 km    (NOAA, 2015), using the SOWFIA project and is
presented in Fig. 4. Data was gathered for period between 1979 and 2009 and the values of Hs
and Tp are recorded every 3 hours.
The histogram and the time-series of average power available in waves were calculated and
shown in Table 2 and Fig. 5, respectively.
As the histogram shows, the largest number of occurrence is 18854, representing 20.81% of
all occurrences and featuring peak period from 6-9 s and significant wave height from 1.5-2
m.



Also, 78.03 % of the waves present significant wave height between 1-2 m.
The minimum and maximum values of significant wave height and peak period recorded are,
respectively 0.59 m and 3.82 m and 2.85s and 22.12 s.

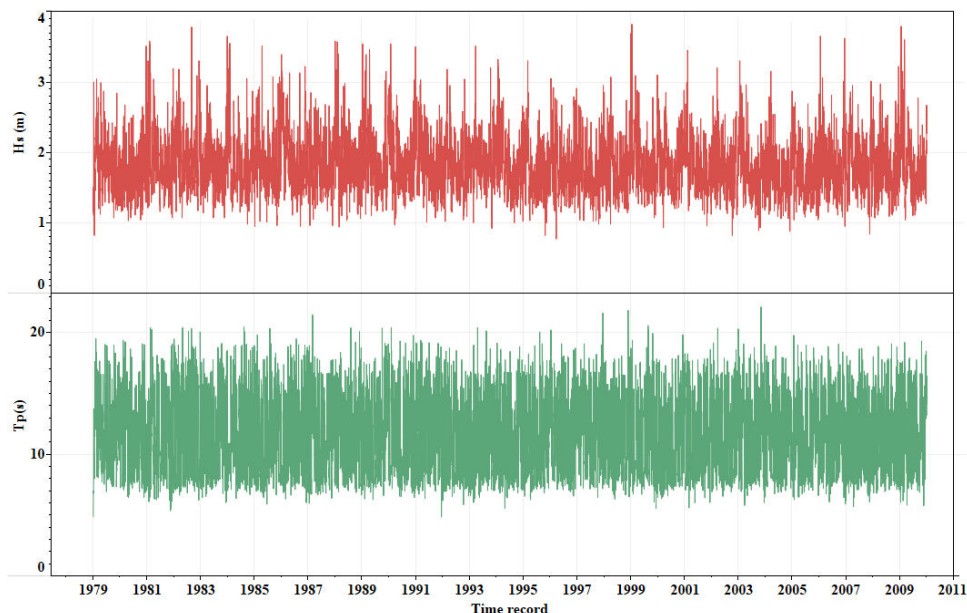

Figure 4. Time-series of significant wave height and peak period between 1979 and 2009
(SOWFIA Project).
The histogram presented in Table 2 shows two local maxima for the peak period, 6–9s and
12–15s, for significant wave height between 1.5–2.5 m. This bimodal distribution indicates a
superposition of two distinct wave regime, the first with origin in a region of a shorter fetch
(smaller period) and the second with origin in a region of longer fetch (longer period). We
suspect that the former is generated during early-year (winter) storms in the North-Atlantic
and the latter during the end-year (autumn) storms in the South-Atlantic.
This is consistent with later findings in this paper that January and December are the most
energetic months and July is the least energetic month.




1    Table 2.Histogram.

| | | Peak Period, Tp[s] | | | | | | | | Occurrence of Hs | %Occurrence of Hs |
|---|---|---|---|---|---|---|---|---|---|---|---|
| | | 1-3 | 3-6 | 6-9 | 9-12 | 12-15 | 15-18 | 18-21 | 21-24 | | |
| Significant Wave Height, Hs [m] | 0-0.5 | 0 | 0 | 0 | 0 | 0 | 0 | 0 | 0 | 0 | 0.00 |
| | 0.5-1 | 1 | 3 | 170 | 427 | 141 | 29 | 6 | 0 | 777 | 0.86 |
| | 1-1.5 | 0 | 572 | 8307 | 9194 | 7288 | 1742 | 127 | 4 | 27234 | 30.07 |
| | 1.5-2 | 0 | 730 | 18854 | 7590 | 12783 | 3315 | 171 | 2 | 43445 | 47.96 |
| | 2-2.5 | 0 | 20 | 8482 | 2072 | 3329 | 1355 | 85 | 0 | 15343 | 16.94 |
| | 2.5-3 | 0 | 0 | 1657 | 731 | 431 | 293 | 25 | 0 | 3137 | 3.46 |
| | 3-3.5 | 0 | 0 | 254 | 219 | 51 | 47 | 7 | 0 | 578 | 0.64 |
| | 3.5-4 | 0 | 0 | 28 | 29 | 3 | 8 | 1 | 0 | 69 | 0.08 |
| | >4 | 0 | 0 | 0 | 0 | 0 | 0 | 0 | 0 | 0 | 0.00 |
| | Occurrence of Tp | 1 | 1325 | 37752 | 20262 | 24026 | 6789 | 422 | 6 | 90583 | 100 |
| | %Occurrence of Tp | 0.00 | 1.46 | 41.68 | 22.37 | 26.52 | 7.49 | 0.47 | 0.01 | 100 | |

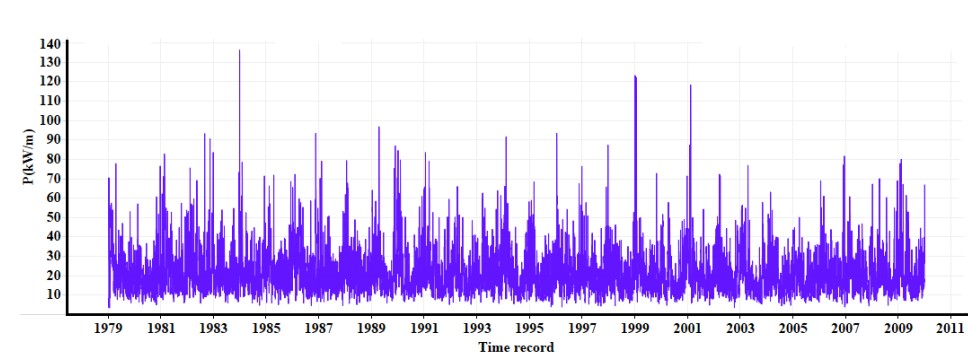

4    Figure 5. Time-series of mean power available on waves between 1979 and 2009.




The curves on Fig. 6 show no clear trend on the time-series of the monthly average power,
over the years. This fact is confirmed by the Mann-Kendal test (Mann, 1945) whose results
are presented at Table 3. The Mann-Kendal test (at 5% level of significance) was done using a
commercially available software (XLSTAT, 2015). The results show that these monthly time-
series can be considered trendless over the years, except for September and October with low
p-values of 3.8% (September) and 1.8% (October) implying a trend.
Table 3. The Mann-Kendall Trend test for monthly average time-series.

| Months | J | F | M | A | M | J | J | A | S | O | N | D |
|---|---|---|---|---|---|---|---|---|---|---|---|---|
| p-values | 0.946 | 0.176 | 0.696 | 0.311 | 0.825 | 0.302 | 0.424 | 0.199 | 0.038 | 0.018 | 0.866 | 0.176 |
| Decision | Without Trend | Without Trend | Without Trend | Without Trend | Without Trend | Without Trend | Without Trend | Without Trend | With Trend | With Trend | Without Trend | Without Trend |

Fig. 7 shows the minimum, average and maximum power available on waves which has been
calculated for each month of the 31 year long record. The graph clearly shows that the most
energetic month is January (23.49 kW/m) and the least energetic month is July (15.04 kW/m).
In fact the average power decays from January to July and increases from July to December
(21.21 kW/m).

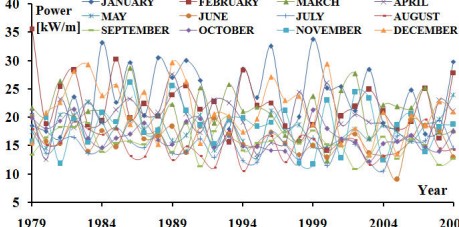

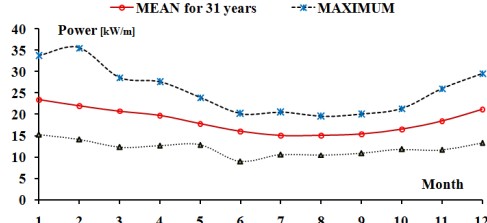

Figure 6. Time-series of monthly average power.

Figure 7. Statistics of monthly average power for 31 years of data.





The increase of the annual average power, between 1979 and 2009 is shown in Fig. 8,
together with its exponential and linear smoothing curves (Hyndmann et al., 2008). The
Dickey-Fuller test helps us to verify if there are upward or downward trends in the time-series
of the annual average power (Kirchgassner and Wolters, 2008). According to this statistical
test (p- values equal to 0.475 and significance level of 5 %), the time-series of the annual
average power is a non-stationary time-series and presents a downward trend, as it is possible
to see by the two smoothing curves . We could not find any plausible explanation for this
downward trend. In this context, it is worth making a forecast of the annual increase of
average power for the next 15 years to see the trend for its predictable values.  To achieve this
goal, it is necessary to calculate the best ARIMA model.

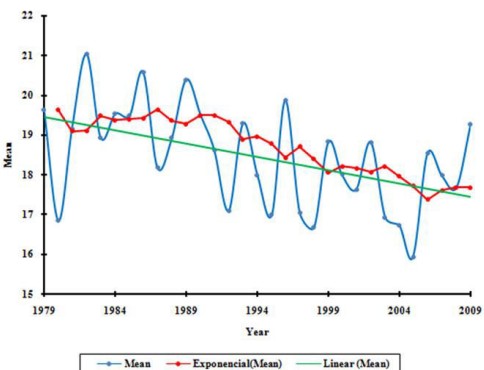

Figure 8. Time-series of annual average power, between 1979 and 2009.
According to  the Dickey-Fuller test, the original time-series of the annual average power is
non stationary. The first difference (P-1) is stationary as its possible to see through the values
of the Autocorrelation Factor (ACF) and of Partial Autocorrelation Factor (PACF), that are
statistically equal to zero, as they are less than 0.35, after Lag = 1 (for ACF) and Lag = 2 (for
PACF). ACF and PACF are two statistical measures that show how the observations in a
time-series are related to each other. Thus, to determine a proper model for a given time-
series, it is necessary to carry out the analysis of these parameters (Frain,1999). Table 4 shows
the values of these quantities under analysis. In the present case, the original time-series is
converted into stationary time-series after the first differencing (d = 1).





Table 4. The ACF and PACF values for P-1 (generated by NCSS10 Software)

**ACF**

| Lag | Correlation | Lag | Correlation | Lag | Correlation | Lag | Correlation |
|---|---|---|---|---|---|---|---|
| 1 | -0.41 | 8 | -0.02 | 15 | -0.02 | 22 | -0.11 |
| 2 | -0.28 | 9 | -0.25 | 16 | -0.17 | 23 | -0.00 |
| 3 | 0.33 | 10 | 0.17 | 17 | 0.15 | 24 | 0.05 |
| 4 | -0.04 | 11 | 0.02 | 18 | 0.09 | 25 | 0.10 |
| 5 | -0.06 | 12 | -0.06 | 19 | -0.27 | 26 | -0.15 |
| 6 | -0.14 | 13 | 0.01 | 20 | 0.17 | 27 | 0.04 |
| 7 | 0.27 | 14 | 0.06 | 21 | 0.01 | 28 | 0.05 |

**PACF**

| Lag | Correlation | Lag | Correlation | Lag | Correlation | Lag | Correlation |
|---|---|---|---|---|---|---|---|
| 1 | -0.41 | 8 | 0.07 | 15 | 0.03 | 22 | -0.03 |
| 2 | -0.55 | 9 | -0.00 | 16 | -0.18 | 23 | -0.10 |
| 3 | -0.13 | 10 | -0.08 | 17 | -0.08 | 24 | -0.05 |
| 4 | -0.08 | 11 | -0.09 | 18 | 0.09 | 25 | -0.04 |
| 5 | 0.08 | 12 | -0.02 | 19 | -0.11 | 26 | 0.04 |
| 6 | -0.27 | 13 | 0.08 | 20 | 0.00 | 27 | 0.09 |
| 7 | 0.09 | 14 | 0.12 | 21 | -0.13 | 28 | 0.02 |

**Significant if |Correlation|> 0.35**

Accornding to Hintze (2007) the value of p is determined from the PACF of the appropriate
differenced time-series. If the PACF cuts off after a few Lags, the last Lag with a large value
would be the estimate for p. Therefore, p is equal to 2 (Table 4). The value of q is estimated,
following the same procedure, using the values of the ACF parameter shown in Table 4. So,
q=1 and, the best ARIMA model to make the forecast is ARIMA (2, 1, 1).
The following table shows the results of the forecast for the annual average power, achieved
using the NCSS Software (NCSSLLS,1981) . According to the forecast, the predicted time-
series of the annual average power oscillates, without any trend, around its average value
(18.2 kW/m). This value is very close to the one presented by Falnes. J. (2007), for the most
tropical waters, where Cape-Verde Island is located.
Table 5. Forecast of Annual Average Power (generated by NCSS10 Software)

| Forecast | | | | |
|---|---|---|---|---|
| Row | Date | Forecast | Lower 95% Limit | Upper 95% Limit |
| 33 | 2011 | 18.05 | 15.69 | 20.41 |
| 34 | 2012 | 17.75 | 15.40 | 20.11 |
| 35 | 2013 | 18.53 | 15.87 | 21.18 |
| 36 | 2014 | 18.33 | 15.55 | 21.11 |
| 37 | 2015 | 18.01 | 15.20 | 20.82 |
| 38 | 2016 | 18.26 | 15.35 | 21.18 |
| 39 | 2017 | 18.32 | 15.28 | 21.35 |
| 40 | 2018 | 18.16 | 15.07 | 21.25 |
| 41 | 2019 | 18.20 | 15.04 | 21.36 |
| 42 | 2020 | 18.27 | 15.01 | 21.52 |
| 43 | 2021 | 18.21 | 14.89 | 21.54 |
| 44 | 2022 | 18.20 | 14.81 | 21.60 |
| 45 | 2023 | 18.24 | 14.77 | 21.70 |
| 46 | 2024 | 18.23 | 14.69 | 21.77 |
| 47 | 2025 | 18.21 | 14.61 | 21.82 |

According to the Portmanteau Test (Hintze, 2007), for a significance level of 5%, the
ARIMA model used to carry out the forecast is adequate, with p-value between 0.179 and
0.641, implying the acceptation of the forecast, as the p-values are higher than the
significance level.
The normality test of Anderson-Darling (Thode, 2002) shows that the annual average power
follows a normal distribution with p-value equal to 51.5% (Fig. 9). As this p-value is higher
than the significance level of 5%, the hypothesis of the normality distribution is accepted.
Fig.9 was generated by Minitab software and represents a summary report of the annual




1 average power time-series. It shows, with a significance level equal to 0.05, the confidence

2 intervals for the annual mean (17.981 kW/m – 18.924 kW/m), for the annual median (17.879

3 kW/m – 19.186 kW/m) and for the annual Standard Deviation (1.028 kW/m – 1.719 kW/m).

4 Fig. 10 shows the normal probability plot for the annual average power. As it is possible to

5 note in this figure, in general, the data follow the normal line. However, some deviataion from

6 this normal line is registed between 16.99 kW/m and 17.09 kW/m.

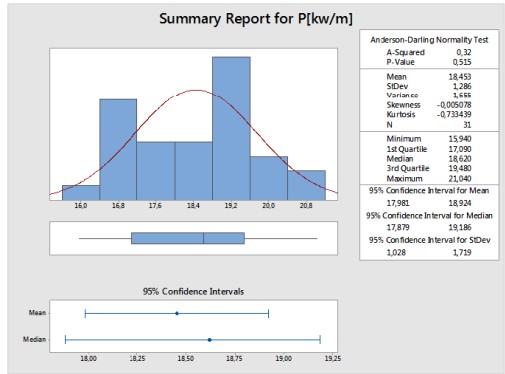

Figure 9. Summary report of annual average
power, between 1979 and 2009.

Figure 10. Normal probability plot.

9 The wave energy resources are stable with COVP less than 0.8, as it is possible to see in Fig.

10 10, which represents the time-series of the annual values of COVP. The MVI parameter

11 shows that the monthly wave energy resources can be considered relatively stable with MVI

12 values less than 1.2 (Fig. 11). This is a very attractive aspect associated with the utilization of

13 wave energy to produce electricity in Cape-Verde since it affects the useful life cycle of ocean

14 wave conversion equipment.

15 Defining a set of samples using all values of the significant wave height, peak period and the

16 average power obtained for each month during the 31 years of data, the confidence intervals

17 for all of these parameters were calculated, using the Minitab software and admitting a

18 significance level of 5% . Before defining the referred confidence intervals the normality tests

19 for all of these parameters were performed. Table 6 summarizes the statistical information



about the normality tests, average values and confidence intervals for each month. The data
are non-normal, as it is possible to see through the values of the A-squared parameter.
According to D'Agostino (1986), the cricital value of the  A-squared parameter, for  a 95%
confidence level, is 0.752. The values of this parameter presented in Table 6 are higher than
this critical value. That is, there is a very strong evidence that the data is non-normal. This
result is confirmed by the p-values that are, in all cases, lower than 0.05 (significance level)
implying the rejection of the normality hypothesis. The Minitab software has a option to
calculate the  confidence intervals for non-normal data. The reseults are presented in Table 6.
Table 6. Monthly statistical reports.

| | Variable | Simple size. N | Anderson-Darling Normality Test | | Mean | StDev | SE Mean | 95% CI |
|---|---|---|---|---|---|---|---|---|
| J | Hs[m] | 7687 | A-Squared: 40.63 | p-value <0.005 | 1.92191 | 0.50899 | 0.00581 | (1.91053; 1.93329) |
| | Tp[s] | 7687 | A-Squared: 170.24 | p-value <0.005 | 10.7142 | 3.1631 | 0.0361 | (10.6435; 10.7849) |
| | P [kW/m] | 7687 | A-Squared: 202.38 | p-value <0.005 | 23.513 | 13.783 | 0.157 | ( 23.205; 23.821) |
| F | Hs [m] | 7008 | A-Squared: 15.53 | p-value <0.005 | 1.87711 | 0.46451 | 0.00555 | (1.86623; 1.88798) |
| | Tp[s] | 7008 | A-Squared: 145.66 | p-value <0.005 | 10.4387 | 3.0192 | 0.0361 | (10.3680;10.5094) |
| | P [kW/m] | 7008 | A-Squared: 208.03 | p-value <0.005 | 21.897 | 12.716 | 0.152 | ( 21.599; 22.195) |
| M | Hs [m] | 7689 | A-Squared: 21.63 | p-value <0.005 | 1.80126 | 0.43902 | 0.00501 | (1.79144; 1.81107) |
| | Tp[s] | 7689 | A-Squared: 70.03 | p-value <0.005 | 10.8515 | 2.8814 | 0.0329 | (10.7871; 10.9159) |
| | P [kW/m] | 7689 | A-Squared: 131.79 | p-value <0.005 | 20.780 | 10.801 | 0.123 | ( 20.538; 21.021) |
| A | Hs[m] | 7440 | A-Squared: 36.30 | p-value <0.005 | 1.80543 | 0.38490 | 0.00446 | (1.79668; 1.81417) |
| | Tp[s] | 7440 | A-Squared: 118.55 | p-value <0.005 | 10.3233 | 2.7986 | 0.0324 | (10.2597; |



| | | | | | |
|---|---|---|---|---|---|
| | | | | | 10.3869) |
| | P[kW/m] | 7440 | A-Squared: 161.64 | p-value <0.005 | 19.763 9.983 0.116 ( 19.536; 19.990) |
| | Hs[m] | 15376 | A-Squared: 29.32 | p-value <0.005 | 1.73386 0.31984 0.00258 (1.72881; 1.73892) |
| M | Tp [s] | 15376 | A-Squared: 491.92 | p-value <0.005 | 10.2287 3.0524 0.0246 (10.1804; 10.2769) |
| | P[kW/m] | 15376 | A-Squared: 258.45 | p-value <0.005 | 17.8068 7.9966 0.0645 (17.6804; 17.9332) |
| | Hs[m] | 14880 | A-Squared: 29.78 | p-value <0.005 | 1.64809 0.30307 0.00248 (1.64322; 1.65296) |
| J | Tp [s] | 14880 | A-Squared: 618.05 | p-value <0.005 | 10.1125 3.0069 0.0246 (10.0642; 10.1608) |
| | P[kW/m] | 14880 | A-Squared: 291.89 | p-value <0.005 | 16.0597 7.4576 0.0611 (15.9399; 16.1795) |
| | Hs[m] | 15376 | A-Squared: 46.52 | p-value <0.005 | 1.59065 0.26830 0.00216 (1.58640; 1.59489) |
| J | Tp [s] | 15376 | A-Squared: 849.41 | p-value <0.005 | 10.1592 2.8717 0.0232 (10.1138; 10.2046) |
| | P[kW/m] | 15376 | A-Squared: 254.08 | p-value <0.005 | 15.0375 6.6470 0.0536 (14.9324; 15.1425) |
| | Hs[m] | 7688 | A-Squared: 27.43 | p-value <0.005 | 1.57631 0.26316 0.00300 (1.57043; 1.58219) |
| A | Tp[s] | 7688 | A-Squared: 337.55 | p-value <0.005 | 10.2906 2.9649 0.0338 (10.2243; 10.3569) |
| | P[kW/m] | 7688 | A-Squared: 174.44 | p-value <0.005 | 15.1119 7.2471 0.0827 (14.9499; 15.2740) |
| | Hs[m] | 7440 | A-Squared: 13.53 | p-value <0.005 | 1.59887 0.27965 0.00324 (1.59251; 1.60522) |
| S | Tp[s] | 7440 | A-Squared: 204.76 | p-value <0.005 | 10.2960 2.8409 0.0329 (10.2315; 10.3606) |
| | P[kW/m] | 7440 | A-Squared: 143.42 | p-value <0.005 | 15.4316 7.0104 0.0813 (15.2723; 15.5910) |
| O | Hs[m] | 7687 | A-Squared: 24.60 | p-value <0.005 | 1.60069 0.33400 0.00381 (1.59322; 1.60816) |





| | | | | | |
|---|---|---|---|---|---|
| | **Tp[s]** | 7687 | A-Squared: 61.21 | p-value <0.005 | 10.8908  2.8969  0.0330 (10.8261; 10.9556) |
| | **P[kW/m]** | 7687 | A-Squared: 188.74 | p-value <0.005 | 16.5502  8.6290  0.0984 (16.3573; 16.7431) |
| | **Hs[m]** | 7440 | A-Squared: 45.13 | p-value <0.005 | 1.65678  0.39347  0.00456 (1.64784; 1.66573) |
| **N** | **Tp[s]** | 7440 | A-Squared: 47.37 | p-value <0.005 | 11.0808  2.9679  0.0344 (11.0133; 11.1482) |
| | **P[kW/m]** | 7440 | A-Squared: 212.76 | p-value <0.005 | 18.439  11.008  0.128 ( 18.189; 18.689) |
| | **Hs[m]** | 7688 | A-Squared: 65.15 | p-value <0.005 | 1.80871  0.45569  0.00520 (1.79852; 1.81890) |
| **D** | **Tp[s]** | 7688 | A-Squared: 110.55 | p-value <0.005 | 10.7810  3.1661  0.0361 (10.7102; 10.8518) |
| | **P[kW/m]** | 7688 | A-Squared: 298.54 | p-value <0.005 | 21.213  13.252  0.151 ( 20.917; 21.509) |

Using the Minitab software, the minimum number of sample points, for average monthly
power, was calculated admitting a 0.85 power factor, a significance level equal to 0.05 and a
value of 3kW/m for margin of error. This margin of error was assumed taking into account
the possibility of completing all measurements in one year. In this context, lower margin of
error implies higher number of sample points. Table 7 show the standard deviations, the
minimum sampling size to guarantee the representativeness of the values of the monthly
average power and, consequently, the number of days to carry out the experiments on the
Natural Caves in order to ensure the correct values of the average power extracted from these
natural infrastructures.
Table 7. Minimum sampling size and the corresponding numbers of days of measurements

| Power Factor: 0.85; Margin of Error: 3 kW/m; Significance level: α = 0.05 | | | |
|---|---|---|---|
| **Months** | **Standard deviation, σ** | **Minimum sampling size, n** | **Numbers of days (for 3 h time step)** |
| **J** | 13.25 | 178 | 23 |
| **F** | 11.01 | 123 | 16 |



| | | | |
|---|---|---|---|
| **M** | 8.63 | 77 | 10 |
| **A** | 7.01 | 51 | 7 |
| **M** | 7.25 | 55 | 7 |
| **J** | 6.65 | 47 | 6 |
| **J** | 7.46 | 58 | 8 |
| **A** | 7.99 | 66 | 9 |
| **S** | 9.98 | 102 | 13 |
| **O** | 10.80 | 119 | 15 |
| **N** | 12.78 | 165 | 21 |
| **D** | 13.78 | 192 | 24 |

**Conclusion**
The most common sea state in Cape-Verde occurs 20.81% of time, featuring peak periods
from 6-9 s and significant wave height from 1.5-2 m. For period between 1979 and 2009,
78.03% of the waves present wave height between 1 and 2 m.
January and December are the most energetic months and July is the least energetic month.
The monthly wave power decreases from January to July and increases again to December.
Through the Coefficient of Variation of Power (COVP) it is possible to conclude that the
wave resource is stable, with COVP between 0.46 and 0.66.
The MVI parameter shows that the wave resource can be considered relatively stable (MVI
<1.2) from monthly average power point of view.
The monthly average time-series is stationary and has no trend over time. The confidence
intervals for all months were calculated using the Minitab software.
The time-series of annual average wave power is non-stationary and presents a visible
attenuation over the years. However, by the use of an appropriate ARIMA model it was
possible to verify that its values oscillate around its average (18.2 kW/m).
The minimum time recording of physical parameters associated with the NC operation are
determined, for each month, under the assumption that the minimum sampling size necessary
to characterize the monthly average power on waves is equal to the minimum sampling size to





characterize the monthly average power emanating from the NC. In this context and for the
Cape-Verde Wave Regime, the minimum sampling size and the corresponding numbers of
days of measurements are given in table 6.
**Acknowledgements**
We are grateful to Jackson Augusto Léger Monteiro for his important contribution to the
realization of this work.

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
