# Peer review of "Statistical Analysis of Wave Energy Resources Available"

_Ocean Science, 2015_

## Referee Comment (RC1) · Anonymous Referee #1 · 12 Feb 2016

This paper presents a statistical analysis of a time-series of significant wave height and peak period recorded in one location near Cape-Verde from 1979 up to 2009. The wave energy power for the 31 years is calculated, and it shows to be decreasing over the years. A characterization of the monthly average wave energy power is also performed. Finally, a forecast of the future availability of the wave energy power is presented.

General comments:

A part for being an interesting/particular area to study, the paper reads more as a technical report than a scientific paper and it is difficult to fully understand with many references to statistical software.

The paper is missing a state-of-the-art section pointing out which are the techniques usually used for quantifying the wave energy power availability and what are the novelties presented in this work. For example are there statistical methods used that go beyond the state-of-the-art of commercially available software? The methodology (and the data used) should be better presented, which would allow to show the scientific novel aspects presented in this paper.

In the presentation of the results there are some points that remain unclear and not fully discussed: 1) the monthly averages do not show any trend in time, but then looking to the annual time series a decreasing trend is present. Should this decreasing trend be present also in the time-series of monthly average power? 2) If a decreasing trend is present analysing the historical data, how can you justify that the future forecast does not show any trend? Is the future forecast not calculated on the basis of the historical data analysed?

The paper can be accepted only after a major revision addressing the above main issues.

Specific comments:

Abstract

Page 1 – Line 11. A reference to the SOWFIA project should be added to allow the reader to have more information on the data used in the paper.

1. Introduction

Adding a state-of-the-art section in the Introduction would help to situate the work and to understand the novel aspects presented. Furthermore, a literature overview of the wave climate in the Cape Verde area, if available, could help to strengthen the paper findings.

Page 5 – Line 14. An additional section "Data" should be added to the manuscript. This will allow the reader to understand better the methodology used. A short description of

the data used is at the beginning of the Results section (page 9 – line 10-14), that can be moved here and further described.

2.1 Average Power

Page 5 – Line 20-21-22. The wave power energy presented in Eq. (1) is valid for any wave spectrum, not only for the Pierson-Moskowits wave spectrum.

Page 6 – Line 1-7. This paragraph is a bit confusing. The significant wave height comes from the integration of wave spectrum too. If you add a "Data" section the description of significant wave height and peak period can go there.

Page 6 – Line 8-15. Have you calculated the wave spectrum using the Pierson-Moskowits parametrization? If not, it is not worth mentioning. After reading this paragraph, it is not clear which is the approach you finally used to calculate the energy period. If you started your analysis from a dataset of significant wave height and peak period, this paragraph can go also in the "Data" section, presenting the starting point of your analysis.

Page 6 – Line 22-24. You could add further discussion and references to other studies about the wave climate in the Cape Verde region. You found that there is a seasonality in the wave power distribution. What happens to the annual trend if one year instead of spanning from January to December (solar year) goes from September to August next year?

Page 7 – Line 1-2. Please explain how data is collected and made available at SOW-FIA, in the "Data" Section.

2.4 Statistical Analysis

Page 8 Line 8. Add a reference to the XLSTAT and Minitab Software.

2.5 Representativeness. . .

Page 8 Line 28. Add a reference to the Minitab Software.

3. Results

Page 9 – Line 10-14. This paragraph can be moved to the "Data" Section. It might be helpful to have a map showing the location of the data collection point and the position of the NC of Cape Verde. Are the waves recorded at 3700 m representative of the wave climate at the NC of Cape Verde? How is the bathymetry in the area?

Page 10 – Figure 4 is not cited and commented in the text. If it adds additional information, please comment it, otherwise remove it.

Page 10 – Line 12-13. Can be more information be added to reinforce your "suspect" ?

Page 11 – Figure 5 is not commented in the text. If it adds additional information, please comment it, otherwise remove it.

Page 12-13 – Figure 6 and Figure 8. If the annual averages come from the average of the monthly averages, it sounds strange that the monthly averages do not show any trend for any of the months, while the annual average time-series show a decreasing trend.

Page 12 – Figure 6 and Figure 7. Is it possible to make them bigger? It might be better to have the same y axis limits for Figure 6 and Figure 8 so we can better compare the annual and monthly trends.

Page 13 – Line 12. The "increase" should be the "decrease".

Page 13 – Line 8-11. It is difficult to get why if you are not able to explain the trend of the historical data "in this context, it is then worth making a forecast".

Page 13 – Figure 8. Add proper label on y axis, missing quantity plotted and units.

Page 14 – Line 10-11. See General Comments.

Page 15 – Table 5. What is "Row"? It might be more readable to see this table as a plot, for example as a continuation of Figure 8.

Page 16 – Figure 10. What is on the axis of Figure 10?

Page 17 – Is it possible to present Table 6 in a Figure? It will help the readability of the manuscript.

Page 19 – Line 7-10. Not clear what you are doing here. Is this an estimate of days of measurements during hypothetical operation of NC in Cape Verde?

Conclusion

Page 20 – Line 12-16. See General Comments.

Page 20 – Line 17-19. Can you elaborate a bit more on this point in the Result section and in the Conclusions.

Page 21- Line 3. "Table 6" should be "Table 7". Looking to Table 7, if I have less numbers of days of measurements during Spring and Summer, what are the implications? Is that important?

Typos:

Page 13 Line 8. Drop the space after "curves"

Page 13 Line 15. Change "its" to "it is"

Page 13 Line 15 "to see" . . . in Table 4?

Page 14 – Line 10. Drop the space after "1981)."

Page 17 Line 8. Change "reseults" to "results".

---

## Referee Comment (RC2) · Anonymous Referee #2 · 16 Mar 2016

The paper describes the statistical analysis of a 30 year time series of wave data for the purpose of evaluating the theoretical resource available for conversion using natural caves in the Cape Verde Islands. This is a useful subject for research, but it is less clear that it could be suitable to report in an Ocean Science journal even if conducted perfectly. It is not obvious to me that it gives any useful new insight to wave climate or surface wave physics (either of which would make it eligible). Furthermore, it is written with an emphasis on statistical analysis and some engineering details, but without very much on the science of surface wind waves and swell. It is also fairly poorly written both in respect to structure and style and grammar. I find it difficult to foresee this paper reaching a suitable standard either in terms of relevance or technical

quality sufficient for publication in Ocean Science.

The introduction starts poorly be giving an absurd value for the global wave resource. Boyle (2004) in fact gives a similar value to most other sources, i.e. an average power of 2TW which equates to 17 500 TWh/year. This error is not relevant to the rest of the paper but makes a very poor impression.

The simplest - but most important - technical deficiency of the paper is its failure to identify the original data sufficiently. We are told the data is obtained from the SOW-FIA project, but as I understand it, SOWFIA was a conduit to gather data rather than responsible for any original data collection. We have a position and dates, but there is no clear indication whether this "data" is from a wave buoy (what type? whose?) or from a wave model (which?). If the data is from a buoy then 30 year time series are sufficiently rare that the data is intrinsically interesting in terms of wave climate. If it is a wave model output then there is little scientific reason to analyze at a single location in isolation, but there might be a case from relevance to Cape Verde Islands. In any case, the data requires an adequate "provenance".

The data is in the form of significant wave heights and peak periods at 3 hour intervals. The authors correctly point out that calculation of the wave power ideally requires an energy period rather than a peak period. The authors discuss methods to estimate an energy period, but fail to make a clear statement of the method that they adopt (e.g. "In this study, we estimate an energy period by Te = 0.86 Tp". Other than this there is a reasonable analysis of the distribution of the data and monthly and annually-averaged time series are constructed.

There are some interesting and wholly credible features in the data. Notably (1) a simple seasonal variation with a single maximum in January and a single minimum in July, (2) an apparent decline in wave power over the 3 decades. I can say these are reasonable observations since the seasonal pattern is very similar to most northern hemisphere wave data and the scientific literature contains many reports and discussion of trends or multi-decadal variation in wave climate. The authors cannot make any such assessment since they appear oblivious to that literature, or at least have not cited any papers on wave climate. I will not criticize not giving a reason for the decline, since this largely eludes all of us, but some reference to causes of longterm variation e.g. behavior of the North Atlantic Oscillation over those decades would have been useful.

I found the discussion of the time series analysis opaque in places, but as I understood, simple analyses of the annually-averaged power suggests a decline, but analysis of each calendar month shows no obvious departure from stationarity in any calendar month. These two statements might appear contradictory, but I think are not surprising given that monthly values will be massively variable. The observation (Figure 7) that the variability is generally greater in the winter months is consistent with other studies that have observed very high inter-annual variability in North Atlantic wave climate in winter months.

I will not go into every statistical analysis in detail, but I am skeptical whether the ARIMA projections are useful. I think it is sufficient to state that we can do no better than "take the past as a guide to the future" and describe the average and variation of wave powers within the 30 year time series.

---

## Author Comment (AC1) · 16 Apr 2016

Report No.3: Answers to the Anonymous Referee #1 The Authors: W. M.L. Monteiro A. J. Sarmento, A.J. Fernandes and J.M. Fernandes 16-04-2016 Dear Anonymous Referee #1, First we want to thank you for all the comments and suggestions you made in relation to our paper. In fact, the improvement of the paper is notable after yours comments. 1. Answer to the general comments (GC) GC1: A part for being an interesting/particular area to study, the paper reads more as a technical report than a scientific paper and it is difficult to fully understand with many references to statistical software A) In fact, several statistical software packages (XLStast, R, Minitab17 and gretl) were used in the study because of the particulars in dealing with the various parameters in the problem.

GC2: In the presentation of the results there are some points that remain unclear and not fully discussed: 1) the monthly averages do not show any trend in time, but then looking to the annual time series a decreasing trend is present. Should this decreasing trend be present also in the time-series of monthly average power? 2) If a decreasing trend is present analysing the historical data, how can you justify that the future forecast does not show any trend? Is the future forecast not calculated on the basis of the historical data analysed?

B) In fact, the monthly average power does not show any trend. However, the inter-annual average power presents a downward trend. If we delve deeper into the analysis of data we find that the trend shown in time-series of the inter-annual average power is illusory and caused by the aggregation effects of data. This situation is now well discussed in the paper. C) The time-series of the predicted values of the inter-annual average power follows the trend of the historical data. A new figure is added to the paper to reinforce this fact. However, a better ARIMA model to predict the future values of the inter-annual average power is found using the R software. The new version of the paper expand on this subject. GC3: The paper is missing a state-of-the-art section pointing out which are the techniques usually used for quantifying the wave energy power availability and what are the novelties presented in this work. For example are there statistical methods used that go beyond the state-of-the-art of commercially available software? The methodology (and the data used) should be better presented, which would allow to show the scientific novel aspects presented in this paper.

A) The procedures and software available for mapping wave energy resources ignore, in general, some important statistical aspects that can lead to errors in wave energy assessment. The outliers that may be present in the time-series of wave data, as a result of a specific event such as extreme storms, could significantly influence the available average wave power. The present study have as novel aspect, the using of the adequate statistical tools to identify possible outliers in time-series of wave data, and the subsequent analyses of their influence in the inter-annual average power calculation. B) Another subject barely mentioned in papers, that can lead to error in the wave energy resources characterization are the effects of data aggregation. The information about the temporal behaviour of the wave data is lost due to the aggregation effects. The present study shows that the aggregation effects may be a real problem that deserves to be analyzed when characterizing wave energy resources. Finally, based on the wave regime characteristics, this paper calculates the time duration necessary to carry on the experiments at Natural Caves aimed to quantify their output power with a minimum sample size that will guarantee its time representativeness. The estimation of the referred time duration is very important as it helps us to evaluate correctly the energetic performance of NCs. In fact, the statistical procedure presented in this paper for quantifying the mentioned time duration can be followed by any researchers to make a better sense of the behaviour of their models of Wave energy devices, through the experimental studies in ocean, or in Lab, using irregular waves. All of these new aspects are now added to the paper. So, we kindly ask you to read a new version of the paper. 2. Answer to the Specific comments (SC)

SC1:Abstract Page 1 – Line 11. A reference to the SOWFIA project should be added to allow the reader to have more information on the data used in the paper.

C) The reference to the SOWFIA project can be found in new section added to the paper. SC2: Introduction Adding a state-of-the-art section in the Introduction would help to situate the work and to understand the novel aspects presented. Furthermore, a literature overview of the wave climate in the Cape Verde area, if available, could help to strengthen the paper findings.

D) A paragraph is added to the data section to highlight the methodology used to evaluate wave energy resources and the novel aspects of the present work. See the answers to the GC3. A state of the art section is added to the paper. In this section we highlight the importance of the present study for Cape-Verde and the real positioning of this country in the context of wave energy utilization. The detection of outliers of in timeseries of the inter-annual average power, their influence in quantifying this quantity, the statistical procedure presented to calculate the time for realization of the experiments on Natural Caves to determining their energetic performance are some aspects of the paper that deserve to be highlighted.

SC3: Page 5 – Line 14. An additional section "Data" should be added to the manuscript. This will allow the reader to understand better the methodology used. A short description of the data used is at the beginning of the Results section (page 9 – line 10-14), that can be moved here and further described.

E) A data section is added to the paper. In this section we make a statement of the SOWFIA project and present a real nature of our data and the inaccuracy associated with them. F) Average Power SC4: Page 5 – Line 20-21-22. The wave power energy presented in Eq. (1) is valid for any wave spectrum, not only for the Pierson-Moskowits wave spectrum. Yes. We agree and new equations for significant wave height calculation is added to clarify this situation. SC5: Page 6 – Line 1-7. This paragraph is a bit confusing. The significant wave height comes from the integration of wave spectrum too. If you add a "Data" section the description of significant wave height and peak period can go there.

Page 6 – Line 8-15. Have you calculated the wave spectrum using the Pierson-Moskowits parametrization? If not, it is not worth mentioning. After reading this paragraph, it is not clear which is the approach you finally used to calculate the energy period. If you started your analysis from a dataset of significant wave height and peak period, this paragraph can go also in the "Data" section, presenting the starting point of your analysis.

Page 6 – Line 22-24. You could add further discussion and references to other studies about the wave climate in the Cape Verde region. You found that there is a seasonality in the wave power distribution. What happens to the annual trend if one year instead of spanning from January to December (solar year) goes from September to August next year? Page 7 – Line 1-2. Please explain how data is collected and made available at SOWFIA, in the "Data" Section.

No, we do not calculated the wave spectrum using the Pierson-Moskowits parametrization. We used an approximation Te=Tp, considered by Hagernam (2001) as we find it very good to make a preliminary study of wave energy resources. In fact, the R software detected seasonality in the monthly average power, if all values of this quantity are aggregate together. But, as mentioned before, the trend in the inter-annual average power is illusory. The data collection by SOWFIA project is presented in the data section. G) Statistical Analysis SC6: Page 8 Line 8. Add a reference to the XLSTAT and Minitab Software. 2.5 Representativeness: Page 8 Line 28. Add a reference to the Minitab Software.

All reference suggested are now added to the paper. H) Results

SC 7: Page 9 – Line 10-14. This paragraph can be moved to the "Data" Section. It might be helpful to have a map showing the location of the data collection point and the position of the NC of Cape Verde. Are the waves recorded at 3700 m representative of the wave climate at the NC of Cape Verde? How is the bathymetry in the area?

We completely agree with your comment and, as suggested, the paragraph is moved to the Data section. Further study on the wave transformation from deep to shallow water must be carried out using information about the local bathymetry. Unfortunately, detailed bathymetric data is available only at some bays and harbour. This lack of information increases the difficulties in producing more realistic results of wave energy resources available at shorelines regions.

SC 8: Page 10 – Line 12-13. Can be more information be added to reinforce your "suspect"?

New information and diagram are added to the paper to better explain the bimodal distribution.

SC9: Page 12-13 – Figure 6 and Figure 8. If the annual averages come from the average of the monthly averages, it sounds strange that the monthly averages do not show any trend for any of the months, while the annual average time-series show a decreasing trend.

The trend observed in the inter-annual average power is results of the aggregation effects of data. This problem is, now, well commented in the text. In fact, this effect appears on the September and October months.

SC10: Page 19 – Line 7-10. Not clear what you are doing here. Is this an estimate of days of measurements during hypothetical operation of NC in Cape Verde?

The quantification of the number of days of measurements of performance parameters of a real Natural Caves operation is one of the important aspects of the present paper. The excitation of real Natural Cave is a time-varying phenomenon. The statistical procedure used in the paper, is based on the wave regime nature to calculate the duration of the experiments to be realized on a real Natural Cave and guarantee the representativeness of the data to be collected during its operation is new. SC11: Page 21- Line 3. "Table 6" should be "Table 7". Looking to Table 7, if I have less numbers of days of measurements during Spring and Summer, what are the implications? Is that important? In fact, the results show that the number of days of measurements is lower in spring (March to May) and summer (June to August). This is due to the lower level dispersion of wave data for these seasons, in comparison with the rest of the months. I) Others suggestion: Typos All others suggestion relative to the graphics aspects of the paper, improvement of figures were taken into account. J) Others news aspects of the paper News paragraphs, graphics section and subsections are added to the paper to clarify many situations of our study.

Please also note the supplement to this comment:
http://www.ocean-sci-discuss.net/os-2015-108/os-2015-108-AC1-supplement.pdf

**Supplement:**

**Statistical Analysis of Wave Energy Resources Available for Conversion at Natural Caves of Cape-Verde Islands.**

**W. M.L. Monteiro[1] A. J. Sarmento[2], A.J. Fernandes[1]and J.M. Fernandes[1]**

[1]{Department of Science and Technology, University of Cape-Verde, Praia, Cape-Verde}

[2]{Lisbon High Technical Institute, Technical University of Lisbon, Lisbon, Portugal}

Correspondence to: W. M.L. Monteiro (wilson.monteiro@docente.unicv.edu.cv)

**Abstract**

Using the time-series of significant wave height and the peak period between 1979 and 2009, generated by SOWFIA - Streamlining of Ocean Wave Farm Impact Assessment, some relevant statistical information about energy content available in ocean waves in Cape-Verde is obtained. The monthly and inter-annual time-series of the average power are analysed and the confidence intervals for their values are defined. Considering all of the 31 years of data, the results show that the most energetic month, from the average power point of view is January (23.49 kW/m) and the least energetic month is July (15.04 kW/m). In fact, the monthly average power decays from January to July and increases from July to December (21.21 kW/m). The inter-annual average power for the 31 years of data exhibits a weak attenuation caused by data aggregation. However, using the moving average smoothing curve it is possible to note that, between 1999 and 2009, the values of this parameter seems to stabilize around 18 kW/m. Using the appropriate Autoregressive Integrated Moving Average (ARIMA) model we verified that the future values of the inter-annual average power tend to oscillate around the same level of average power (18 kW/m).The outliers present in time-series of annual average power were identified and their influence in the value of inter-annual average power was quantified. Removing outliers from the annual time-series of power caused a maximum relative attenuation in the values of the inter-annual average power between 1.85 and 13%. Through the Coefficient of Variation of Power (COVP), obtained by dividing the standard deviation of the power time-series by the average power, it is possible to conclude that the wave resource is stable, with COVP between 0.46 and 0.66. The values of the Monthly Variation Index (MVI), the maximum range of the monthly mean wave power relative to the yearly mean level, show that the resource is relatively stable, with MVI < 1.2. The present work calculates the deep water power available for the Natural Caves (NCs) in Cape Verde Islands, through a rigorous analysis of the wave climate that excites them. The minimum sampling size and the corresponding numbers of days of measurements per month are also estimated. The results show that the number of days of measurements is lower in spring (March to May) and summer (June to August). This is due to the lower level dispersion of wave data for these seasons, in comparison with the rest of the months.

**1   Introduction**

Ocean waves constitute one of the renewable sources of energy that are gradually entering the market of clean and sustainable energy worldwide. The global theoretical energy from ocean wave is estimated in 17500 TWh/year (Boyle, 2004). Many countries around world have been investing on this natural resource to produce useful and sustainable energy. Portugal (Pelamis and Pico Plant Projects.), Australia (CETO and OCEANLIX projects), France (SEAREV project), UK (OYSTER WEC and Limpet projects) and Holland (AWS project) are examples of some of countries that have recognized the feasibility of harvesting this source of energy (ABP, 2004). According to the International Renewable Energy Agency (Monford et al., 2014), around 64 % of the Wave Energy Converters (WECs) has been projected for offshore application and 36% for near-shore and onshore operation. Some full-scale operational tests have been realized. These include the OYSTER device from Aquamarine Power, the Wave Roller from AW-Energy, Pelamis P2 from the Pelamis Wave Power, the Seabased and the Sea-Tricity devices. Magagna (2011) has identified, in 2011, over 100 wave energy developers. Yet, EMEC (2014) has listed 170 wave energy developers worldwide. About 45% of the wave energy developers are based in or are currently developing projects in the European Union (EU) regions. The global installed capacity of wave energy remains low and the technologies are still at an advanced R&D stage. Just a few machines have sustained long operational hours, such as the Aquamarine OYSTER (>20000 hours) and Pelamis (cumulative > 10000 hours) (Scottish Renewable, 2014). The growth of the wave energy sector is lower than expected and this fact may affect the confidence of investors in this area. Success in attracting future Original Equipment Manufacturer investments will depend on the capacity of the developers in improving performance, reducing cost and validating wave energy technologies. The long-term global wave energy is expected to become cost competitive and provide an alternative to other Renewable Energy Sources and conventional energy resources. Through a review of the existing data available, the different cost components in the Capital Expenditure (CAPEX) estimate for wave energy extraction have been identified as follows (Table 1):

Table1. Costs components estimate for wave energy extraction (JRC, 2014).

| Civil and Structural costs | 38% |
| Mechanical Equipment costs | 42% |
| Electrical and I&C supply and installation costs | 8% |
| Project indirect cost | 7% |
| Development cost | 5% |

Thus, the main components of CAPEX are mechanical equipment, civil and structural costs. In this context, the developers of wave energy technologies must undertake efforts and strategies aimed at reducing the two above mentioned costs and the risks associated with the operation of these equipment out-shore or close to shore.

**1.1 Natural Caves**

Natural Caves (NC) are caverns formed naturally under the rocky shorelines, inside of which there is an air layer (Fig.1). This air layer acts like an air pump against the cave ceiling, as the wave enters and exits these natural infrastructures, forcing the compressed air to go in and out of the NC, through surface holes at the top of the cave. Fig. 2 shows NCs with one and two holes.

[Figure]

[Figure]

Figure 1. Activity in a Natural Cave.    Figure 2. Activity in a NCs with two holes.

The principles of NCs operations are similar to the man-made Oscillating Water Column
device, projected for onshore application.

Justification for using the NCs for wave energy conversion is a possible cost reduction on the
Civil and Structural cost components, which, as mentioned before, are the most significant
costs associated with building wave energy devices to produce electricity. Furthermore, the
risk of the device collapsing is minimized, by taking advantage of the sturdiness of the natural
rocky structure, time tested by the waves and storms.

To evaluate the potential of NCs for electricity production, it is necessary to estimate its
output power. To do this, a set of experiments aimed at determining the values of some
important physics parameters of NCs operations need to be conducted. Monteiro and
Sarmento (2015) carried out the analytical modelling of the NCs operations as a function of
their functioning physical parameters. The present study is part of a deeper work aimed at
quantifying the output power of NCs and to project an adequate power take-off system to be
adapted on their holes, for energy extraction.

Since the excitation waves are irregular, non-linear and non-stationary phenomenon it is very
important to determine beforehand the sampling size, i.e. how long it takes to carry out the
experiments (number of days of measurements) on a NCs, in order to guarantee the time
representativeness of its output power. To achieve this goal, some statistical analysis has to be
carried on the wave energy input regime.

**1.2   Wave Energy in Cape-Verde: the state of the art**

Cape-Verde is an archipelago of ten islands in the Atlantic Ocean, off the West Coast of Africa, with roughly half million people. The country is totally dependent on oil to produce electricity, having one of the most expensive cost of electricity in Africa, around 0.28 Euro/kWh (Electra, 2012) versus 0.17 Euro/kWh (Senelec, 2015) at Senegal, a continental neighbour. Some investments were made by the Government of Cape-Verde aimed to introducing renewable sources of energy in the country, mainly solar and wind energy. The Renewable Energy Plan for Cape-Verde (ERPCV)  has defined an ambitious goal of achieving 50% of Renewable Energy penetration in the country by 2020 (GESTO, 2011). As a results of the ERPCV, there are in the country  four wind energy farms with a total annual production between 80 and 110 GWh and two solar energy farms with 7.5 MWp (MWp-Mega Watt Peak) (GESTO, 2011). In 1999, some research projects on ocean energy were initiated in the country, directed at Ocean Thermal Energy Conversion (OTEC) system and WaveStar technology (Wave energy). Unfortunately, these projects did not produced any visible results since they lacked a institutional framework on which to develop.

Because of its insular nature, most of Cape-Verde's economic activities (around 90%) are concentrated on coastal areas (Carvalho, 2013). In this context, it makes sense to use wave energy for producing electricity locally. A clear alternative is harvesting the energy from ocean waves. The evaluation of the wave energy resources and the feasibility study associated with its utilization in Cape-Verde need to be assessed in more detail. Before 2009, some pilot projects for wave energy conversion at southern of Santiago Island were conceived but never implemented and worst yet, never went beyond pre-feasibility studies (DGE, 2009). In 2009 an attempt was made to deploy the WaveStar device, developed by Danish company WaveStar Energy, at Sal Island. This project forecasted the wave energy resources in some regions around the Island, using measurements gathered by a wave buoy installed in place. Unfortunately, the project failed to achieve its goals and the buoy was abandoned at the Instituto de Meteorologia e Geofisica de Cabo-Verde, in Sal Island (DGE, 2009). In 2011, GESTO Energy, a Portuguese company, carried out an evaluation of the wave resources in Cape-Verde based on eleven years of data produced by meteorological wave model worldwide. The data of direction, period and significant wave height were characterized and the values of these parameters were used for calculate the offshore annual average wave power (GESTO, 2011). According to this study, the islands that present the best potential for wave energy exploration are Sal, S. Antão, S. Vicente and Boa Vista. In fact, four projects for offshore wave energy conversion based on the Pelamis technology were proposed for these islands (GESTO, 2011): Sal (3.7 MW), S. Antão (3.7 MW), S.Vicente (3.7 MW) and Boavista (3.5MW). The study was commissioned by the Ministry of Turism, Industry and Energy of Cape-Verde and, unfortunately, the scientific results of the study are unknown since it was never published in any scientific journal or conference proceedings.

As there are no scientific data available on wave energy resources for Cape Verde Islands, the present work brings to light the real potential for wave energy harvesting and constitutes a significant contribution to authorities on which to base any decision about forthcoming investments on wave energy conversion. However, more detailed information will be needed in order to accurately validate the result of this study, as will be shown in next section.

**2   Data**

Knowledge of wave energy resource at a certain location is required by developers of Wave Energy Converters projects in order to allow them to select the most favourable sites for achieving optimal power capture and economic performance from their devices. Three main categories of data are available for wave energy resources assessment: In-situ measurements (buoy, pressure transducers, wave staff, ship-borne wave recorders), remote sensing (satellite radar Altimetry RA, Synthetic Aperture Radar SAR, Marine Wave Radar), numerical models for deep-water (WAM and WaveWatch 3) and for shallow-water (SWAN, TOMAWAC and MIKE21).

SOWFIA-Streamlining of Ocean Wave Farm Impact Assessment is an EU Intelligent Energy Europe Project with the goal of sharing and consolidating pan-European experience and best practices for consenting processes and environmental and socio-economic impact assessment (IA) for offshore wave energy conversion developments (Mora-Figueroa et al., 2011). This project brings together ten partners across eight EU Member States actively involved in planned wave farm test centres and aims at providing recommendations for streamlining of IA approval processes with the purpose of removing legal, environmental and socio-economic barriers associated with development of the wave energy farms. The SOWFIA project uses data obtained from direct measurements (wave buoy) of the wave climate, carried out at the seven European wave energy test centres, through the Data Management Platform (DMP) tool. DMP is an interactive tool designed to assist in the decision making process, providing information on different wave energy monitoring activities at different test centres and allowing direct visualization and downloading of relevant data. The DMP is publically available on the SOWFIA website. The seven European test centres involved in the SOWFIA project are the AMETTS (Ireland), BIMEP (Spain), Lysekil (Sweden), Ocean Plug (Portugal), SEAREV (France), Wave Hub (United Kingdom) and EMEC (Scotland) (Mora-Figueroa et al., 2011).

For others regions of the ocean, where there are no in situ data measurements, the SOWFIA project uses data produced by WaveWatch 3 (WW3) wave model. The WW3 is phase-average model that solves the spectral action density balance equation for wavenumber-direction spectra. The Governing equation includes refraction due to the temporal and spatial variation of the mean water depth and current. The source terms include nonlinear interactions, dissipation due to the white capping, bottom friction, wind wave growth and decay (Tolman, 1999). An important constraint of the formulation of the WW3 is that the parameterizations of the physical process included in the model do not address conditions where the waves are strongly depth-limited. This constraint implies that the model is generally applied on spatial scales between 20 and 100 km outside of surf zone (Tolman, 1999).

Like other sources of renewable energy, the nature of ocean waves is complex and impossible to be predicted precisely. The data produced by WW3 model must be, wherever possible, calibrated with in situ measurements using wave buoy or altimeter data. Both calibrations of the wave data and the estimation of the confidence bounds are made difficult by the complex structure of errors in the model data. Error in parameters from wave model show nonlinear dependence of variety of factors, seasonal and inter-annual changes in bias and short-term temporal correlation (Mackay et al., 2010). To assess the uncertainty associated with the estimation of energy yield from a wave energy converter (WEC), Mackay et al (2010) use two hindcasts from European Marine Energy Centre in Orkney. These hindcasts are produced by WAM (Komen et al., 1994) and WW3 wave models and calibrated using a Datawell Directional Waverider buoy moored in 50 m water depth at the EMEC site. The study show that before wave data calibration, the estimation of the long-term mean WEC power from the two hindcasts differ by around 20%. After calibration this difference is reduced to 5%.

Data produced by WW3 through the SOWFIA project is used to evaluate the wave energy resources at Cape-Verde. The data was gathered for period between 1979 and 2009, at coordinates 16ºN-24ºW, approximately at centre of the archipelago, where the water depth is around 3.7 km (NOAA, 2015). The WW3 produced information about the significant wave height ($H_s$), peak period ($T_p$), peak direction ($D_p$) and wind velocity, every 3 hours. The data generated by a wave model, should have been calibrated against data collected in situ, but unfortunately, there is no in-situ calibration buoy in the region. Another factor which introduces some inaccuracy in the data, is the shadow effect caused by the own presence of the islands. According to Ponce de Léon et al. (2010) the shadow effect is not taken into account in wave models and can introduce inaccuracy in wave data, especially at the location where the wave regime is characterized by low values of $H_s$. Another important aspect that deserves mentioning is the location of the Natural Caves (inshore) relative to the location of the data acquisition (offshore). Further study on the wave transformation from deep to shallow water must be carried out using information about the local bathymetry. Unfortunately, detailed bathymetric data is only available for some bays and harbours. Therefore, further approximations of coastal bathymetry must be made in order to obtain a more realistic result of wave energy resources available at shorelines regions.

The procedures and software available for mapping wave energy resources ignore, in general, some important statistical aspects that can lead to errors in wave energy assessment. The outliers that may be present in the time-series of wave data, as a result of a specific event such as extreme storms, could significantly influence the available average wave power. Yet, as the experimental study carried out by Mendes and Monteiro (2007) shows, some inshore WEC such as the OWC device, present serious handicaps when operating in high waves since these waves produce significant hydrodynamic loss associated with the interaction between waves and the caisson structure. Thus, the present study introduces a novelty by using adequate statistical tools to identify possible outliers in time-series of wave data, and the subsequent analyses of their influence in the inter-annual average power calculation. Another subject barely mentioned in papers, that can lead to error in the wave energy resources characterization are the effects of data aggregation. The information about the temporal behaviour of the wave data are lost due to the aggregation effects. The present study shows that the aggregation effects may be a real problem that deserves to be taken into account when characterizing wave energy resources. Finally, based on the wave regime characteristics, this paper calculates the time duration necessary to carry on the experiments at Natural Caves aimed to quantify their output power with a minimum sample size that will guarantee its time representativeness. The estimation of the time duration is very important as it helps evaluate correctly the energetic performance of NCs. In fact, the statistical procedure presented in this paper for quantifying the time duration can be followed by other researchers to better understand the behaviour of their models of wave energy devices.

**3   Methodology**

[revised manuscript text omitted]

**4   Results**

Table 2 shows the histogram of wave regime, where 78.03 % of the waves have significant wave height of 1-2 m and 20.81% of all occurrences feature peak period from 6-9 s and significant wave height from 1.5-2 m. The minimum and maximum values of significant wave height and peak period recorded are, respectively 0.59 m and 3.82 m and 2.85s and 22.12 s. Yet, the histogram presented in Table 2 shows two local maxima for the peak period 6 to 9s and 12 to 15s, for significant wave height between 1.5 and 2.5 m. This bimodal distribution indicates a superposition of two distinct wave regimes, the first with origin in a region of a shorter fetch (smaller period) and the second with origin in a region of longer fetch (longer period). Fig.4 shows the wave rose diagram obtained for these two wave regimes. The first diagram (A) represents the predominant direction for peak periods between 6s and 9s. These waves are generated by the predominant winds, blowing constantly throughout the year, from NNE direction. Since there is not enough fetch length between Cape Verde Islands and the African continent (600 Km), the wave regime do not fully develop and remains with a peak period between 6 and 9 seconds.

The second diagram (B) represents the predominant direction for peak periods between 12s and 15s and shows a superposition of waves from two origins:

- NNW waves which are generated during early-year winter storms in the North-Atlantic

• SSW waves which are in turn generated during the end-year autumn storms in the

South-Atlantic.

Both regions have sufficient fetch length to fully develop the wave regime, and it may be possible to observe outliers of 17s – 18s generated in South-Atlantic.

This is consistent with later findings in this paper that January and December are the most energetic months and July is the least energetic month.

[Figure]

|  | A |  | B |
|---|---|---|---|

Figure 4. Wind rose for two local maxima characterized by the peaks periods between 6 and 9 s (A) and between 12 and 15 s (B).

Table 2. Histogram.

| | | Peak Period, Tp[s] | | | | | | | | Occurrence of Hs | %Occurrence of Hs |
|---|---|---|---|---|---|---|---|---|---|---|---|
| | | 1-3 | 3-6 | 6-9 | 9-12 | 12-15 | 15-18 | 18-21 | 21-24 | | |
| **Significant** | 0-0.5 | 0 | 0 | 0 | 0 | 0 | 0 | 0 | 0 | 0 | 0.00 |
| | 0.5-1 | 1 | 3 | 170 | 427 | 141 | 29 | 6 | 0 | 777 | 0.86 |

| | | | | | | | | | | | |
|---|---|---|---|---|---|---|---|---|---|---|---|
| **1-1.5** | 0 | **572** | **8307** | **9194** | **7288** | **1742** | **127** | **4** | 27234 | 30.07 |
| **1.5-2** | 0 | **730** | **18854** | **7590** | **12783** | **3315** | **171** | **2** | 43445 | 47.96 |
| **2-2.5** | 0 | **20** | **8482** | **2072** | **3329** | **1355** | **85** | 0 | 15343 | 16.94 |
| **2.5-3** | 0 | 0 | **1657** | **731** | **431** | **293** | **25** | 0 | 3137 | 3.46 |
| **3-3.5** | 0 | 0 | **254** | **219** | **51** | **47** | **7** | 0 | 578 | 0.64 |
| **3.5-4** | 0 | 0 | **28** | **29** | **3** | **8** | **1** | 0 | 69 | 0.08 |
| **>4** | 0 | 0 | 0 | 0 | 0 | 0 | 0 | 0 | 0 | 0.00 |
| **Occurrence of Tp** | **1** | **1325** | **37752** | **20262** | **24026** | **6789** | **422** | **6** | **90583** | 100 |
| **%Occurrence of Tp** | **0.00** | **1.46** | **41.68** | **22.37** | **26.52** | **7.49** | **0.47** | **0.01** | **100** | |

The curves on Fig. 5 show no clear trend on the time-series of the monthly average power, over the years. This fact is confirmed by the Mann-Kendal test (Mann, 1945) whose results are presented at Table 3. The Mann-Kendal test (at 5% level of significance) was done using commercially available software (XLSTAT, 2015). The results show that these monthly time-series can be considered trendless over the years, except for September and October with low p-values of 3.8% (September) and 1.8% (October). However, these trends should be the results of data aggregation error that will be reported in more detail later in this work.

Table 3.The Mann-Kendall Trend test for monthly average time-series.

| Months | J | F | M | A | M | J | J | A | S | O | N | D |
|---|---|---|---|---|---|---|---|---|---|---|---|---|
| **p-values** | 0.946 | 0.176 | 0.696 | 0.311 | 0.825 | 0.302 | 0.424 | 0.199 | 0.038 | 0.018 | 0.866 | 0.176 |
| **Decision** | Without Trend | Without Trend | Without Trend | Without Trend | Without Trend | Without Trend | Without Trend | Without Trend | With Trend | WithTrend | Without Trend | Without Trend |

[Figure]

Figure 5.Time-series of monthly average power.

Fig. 6 shows the minimum, average and maximum power available on waves which has been calculated for each month of the 31 year long record. The graph clearly shows that the most energetic month is January (23.49 kW/m) and the least energetic month is July (15.04 kW/m). In fact, the average power decays from January to July and increases from July to December (21.21 kW/m). The curves presented in Fig.6 show the same behaviour as those obtained by Mackay et.al (2010), for a region off the north coast of Scotland.

[Figure]

Figure 6. Statistics of monthly average power for 31 years of data.

Fig.7 shows the inter-annual average values of significant wave height (Hsav) and peak period (Tpav), together with their moving average smoothing cuves (MASC) of seven periods

7 (Hyndmann et al., 2008). According to this figure, the maximum and minimum inter-annual average values of the significant wave height are equal to 1.83 m and 1.61 m, recorded, respectively, in the years 1986/1997 and 2005. After 1990, the inter-annual average values of the significant wave height shows a rapid decay until 1998, after which it shows a quasi- constant behavior around 1.68 m, until 2009. The inter-annual average values of the peak period has its maximum value of 11.00 s in 1997 and  minimum of 9.91 s, in 2007. The values of this parameter show a downward trend until 1994. Between 1994 and 2002 the inter-annual values of the peak period have an upward trend and after 2002 it shows a quasi-constant behaviour, around 10.5 s. Similar trend is observed for  the significant wave height parameter.

[Figure]

Figure 7. Inter-annual average values of significant wave height and peak period.

The inter-annual average values of power available in waves are shown in Fig.8. The maximum and minimum values of this parameter are 21.04 kW/m and 15.94kW/m attained, respectively, in 1982 and 2005.  The inter-annual average values of power show a downward trend until 1998, after which it exhibits a quasi-constant behavior around 18 kW/m, until
2009.

[Figure]

Figure 8. Time-Series of inter-annual average power.

For a more in depth analysis of the trend of the inter-annual average power time-series the
Augmented Dikey-Fuller (ADF) trend test was used. As the Fig.8 shows, the time-series of
the inter-annual average power shows an initial downward trend and a constant, as its values
oscillate around a nonzero constant. Thus, in the ADF trend test we assume that there is a
constant and a trend. Another aspect associated with the utilization of the ADF test is the
calculation of the optimum Lag length. To do this, the calculation of the maximum Lag length
($Lag_{max}$) is necessary. This can be done using the equation $Lag_{max} = int\left\{12\left(^{T}\!/_{100}\right)^{^{1}\!/_{4}}\right\}$,
suggested by Schwert (1989). In this equation, "int" means that we must accept the integer
parts of the results produced by the equation and T is the dimension of sample. For our case
study, T = 31 (for inter-annual time-series) observations and, therefore, the $Lag_{max} = 9$.
Using the function "Var Lag" in gretl software, it is possible to calculate, automatically, the
optimal Lag length, according to Akaike Information Criterion (AIC), Bayesian Information
Criterion (BIC) and Hannan-Quinn Information Criterion (HQC) (Komm, 2015). The results
produced by this procedure are presented in the following Table.

Table 4. The AIC, BIC and HQC values as a function of Lag length, for inter-annual average power time-series.

| Lags | 1 | 2 | 3 | 4 | 5 | 6 | 7 | 8 | 9 |
|------|------|------|------|------|------|------|------|------|------|
| AIC | 3.23* | 3.27 | 3.29 | 3.33 | 3.42 | 3.48 | 3.41 | 3.50 | 3.49 |
| BIC | 3.38* | 3.47 | 3.54 | 3.63 | 3.76 | 3.88 | 3.86 | 3.99 | 4.03 |
| HQC | 3.27* | 3.32 | 3.35 | 3.40 | 3.50 | 3.57 | 3.51 | 3.62 | 3.62 |

In Table 4, "*" means the best Lag length. Thus, the optimum Lag length is, then, Lag $= 1$, according to all the criteria mentioned before. Using the gretl software for this optimal value of the Lag length and the for the assumptions of existence of constant and trend, as mentioned before, the ADF trend test produces a very low $p - value = 0.0001$, in comparison with the level of significance ($\alpha = 0.05$) used to perform the test. Therefore, the null hypotheses of non-stationarity must be rejected. Thus, the time-series present a deterministic trend that is, the inter-annual average time-series of power is a trend-stationary process (Stadnitski, 2009).

This kind of trends is caused by a moving average component that is an explicit function of time. To better understand the nature of the trend, exhibit by the values of inter-annual average power, it would be worth to carry on the trend test of the original time-series of power. Fig.9 shows the original time-series of power, between 1979 and 2009 calculated for each 3h. Analyzing this figure becomes clear that the values of the power oscillate around a constant different from zero and there is no clear evidence of trend in the value of the referred parameter.

[Figure]

Figure 9. The original time-series of annual power, between 1979 and 2009.

The optimum Lag length is calculated and is equal to Lag = 21 (AIC), Lag = 7 BIC and Lag = 9 (HQC). These different results for optimum Lag-length could be associated with the heterogeneity of our data. The ADF test was carried out for all three Lag lengths and all of them produced a rejection of null hypothesis of non-stationarity. So, the original time-series of power is stationary around a constant mean. These results lead us to conclude that the initial trend shown in the values of the inter-annual time-series of power could have been caused by two factors: the effects of aggregation (Clark et al., 1976) and the existence of the Outliers, defined as an observation in data set which appears to be inconsistent with remainder of that set of data (Johnson, 1992). These outliers could affect significantly the mean values of power. According to Clark et al. (1976), aggregation problem can be defined as the information loss which occurs in the substitution of aggregate, or macro-level, data for individual, or micro-level, data. This undesirable effect reduces the variability of data. In fact, aggregating the values of the power into its inter-annual average values produce, in our case, a reduction of standard deviation parameter from 10.39 kW/m, in original time-series, to 1.29 kW/m, in inter-annual time-series. This corresponding to a dramatic reduction of 87.5% of standard deviation in comparison with the value of this quantity for the original time-series, and it could introduce a high level of error associated with the aggregation effects.

To analyze the implications of the outliers in our results, they were identified, through the box plot method (Ben-Gal, 2005), using, for the present study, the R software, and subsequently removed from the time-series. The numbers of the outliers found, in this way, for each time-series of the annual average power, are presented in Table 5.

Table 5. Numbers of outliers present in each annual time-series of power

| Year | 1979 | 1980 | 1981 | 1982 | 1983 | 1984 | 1985 | 1986 | 1987 | 1988 | 1989 | 1990 | 1991 | 1992 | 1993 | 1994 | 1995 | 1996 | 1997 | 1998 | 1999 | 2000 | 2001 | 2002 | 2003 | 2004 | 2005 | 2006 | 2007 | 2008 | 2009 |
|---|---|---|---|---|---|---|---|---|---|---|---|---|---|---|---|---|---|---|---|---|---|---|---|---|---|---|---|---|---|---|---|
| Numbers of Outliers | 130 | 132 | 151 | 120 | 111 | 146 | 102 | 107 | 104 | 92 | 155 | 125 | 110 | 124 | 128 | 114 | 95 | 149 | 149 | 78 | 134 | 106 | 124 | 118 | 187 | 95 | 31 | 134 | 101 | 108 | 180 |

As we mentioned before, all outliers are removed from the time-series of annual average power. Further, the time-series of the inter-annual average power is, then, calculated, and the results are plotted in Fig.10, together with the correspondent inter-annual average power including outliers. As the referred figure shows, the trend-stationarity process persists in the time-series of the inter-annual average power even when removing the outliers. That is, the trend is not caused by the influence of outliers. But, they introduce a slightly relative variation in the values of inter-annual average power, which maximum varies between 1.85% and 13%.

However, at sites of extreme stroms, severe outliers may appear.  In this context, it is worth analyzing the influence of these severe outliers in the context of wave energy resource characterization. Now, it is clear that the trend of the inter-annual average power is a result of the effect of data aggregation.

[Figure]

Figure 10. The time-series of inter-annual average power, with and without outliers.

To estimate the future behavoir of the values of the inter-annual average power a forecast for the next 10 years is performed. For this purpose, it is necessary to calculate the best ARIMA

model.

According to  the ADF trend test, the original time-series of the inter-annual average power is trend-stationary. The first difference (P-1) is stationary as it is possible to see through the values of the Autocorrelation Factor (ACF) and of Partial Autocorrelation Factor (PACF)

presented in Fig.11, generated by gretl software. In fact, as shown in Fig. 11, the values of these parametersare statistically equal to zero, as they are less than 0.35, after Lag = 1 (for

ACF) and Lag = 2 (for PACF). ACF and PACF are two statistical measures that show how the observations in a time-series are related to each other. Thus, to determine a proper model for  a  given  time-series,  it  is  necessary  to  carry  out  the  analysis  of  these  parameters (Frain,1999). In the present case, the original time-series is converted into stationary time- series after the first differencing (d = 1).

[Figure]

Figure 11. The values of ACF and PACF parameters for the inter-annual average power time- seires.

Accornding to Hintze (2007) the value of p is determined from the PACF of the appropriate differenced time-series. If the PACF cuts off after a few Lags, the last Lag with a large value would be the estimate for p. Therefore, p is equal to 2 (Fig.11). The value of q is estimated, following the same procedure, using the values of the ACF parameter shown in Fig.11. So, q=1 and, the best ARIMA model to make the forecast is ARIMA (2, 1, 1).  However, using the R software it is possible to generate automatically, the best ARIMA model to make a forecast of a time-series. For our inter-annual average power time-series the R software produce the ARIMA (2,1,0). According to AIC and HQC criteria the ARIMA model generated by R software is better than ARIMA (2,1,1). In fact, the ARIMA (2,1,0) led to the lower values of AIC (103.78) and HQC (105.57) in comparision to which presented by

ARIMA (2,1,1) that were, respectively, 104.83 for AIC and 107.07 for HQC.  Thus, in the present study, the forecast was made using the best ARIMA model, that is the ARIMA

(2,1,0).

Fig.12 shows the results of the forecast for the inter-annual average power, achieved using the gretl software. As Fig.12 shows, the predicted time-series follows the observed time-series and produced a residual values that oscillate around zero (Fig.13), which shows that the predicted values tend to adjust to the observed values. According to the forecast, the predicted time-series of the inter-annual average power seems to oscillates, without any trend, around of

18 kW/m, as was previously predicted using the moving average smoothing curve. This value is very close to the one calculated by Falnes. J. (2007), for tropical regions, similar to

Cape-Verde Island.

[Figure]

Figure12. Forecast of Annual Average Power (generated by R software).

[Figure]

Figure13. Residual values of the Forecasted inter-annual Average Power time-series (generated by R software).

The normality test of Anderson-Darling (Thode, 2002) shows that the inter-annual average
power follows a normal distribution with p-value equal to 51.5% (Fig. 14). As this p-value is
higher than the significance level of 5%, the hypothesis of the normality distribution is
accepted. Fig.14 was generated by Minitab software and represents a summary report of the
inter-annual average power time-series. It shows, with a significance level equal to 0.05, the
confidence intervals for the inter-annual mean (17.981 kW/m – 18.924 kW/m), for the inter-
annual median (17.879 kW/m – 19.186 kW/m) and for the inter-annual Standard Deviation
(1.028 kW/m – 1.719 kW/m). Fig. 15 shows the normal probability plot for the inter-annual
average power. As it is possible to note in this figure, in general, the data follow the normal
line. However, some deviation from this normal line is registed between 16.99 kW/m and
17.09 kW/m.

[Figure]

Figure 14. Summary report of inter-annual
average power, between 1979 and 2009.

Figure 15. Normal probability plot.

The wave energy resources in Cape-Verde are stable with COVP less than 0.8, as it is
possible to see in Fig. 16-A, which represents the time-series of the inter-annual values of
COVP. The MVI parameter shows that the monthly wave energy resources can be considered
relatively stable with MVI values less than 1.2 (Fig. 16-B). This is a very attractive aspect
associated with the utilization of wave energy to produce electricity in Cape-Verde, since it
affects the useful life cycle of ocean wave conversion equipment.

[Figure]

Figure 16. Temporal variability of wave resources. A – Coefficient of Variation of Power; B – Monthly Variation Index.

Defining a set of samples using all values of the significant wave height, peak period and the
average power obtained for each month during the 31 years of data, the confidence intervals
for all of these parameters were calculated, using the Minitab software and admitting a
significance level of 5%. Before defining the referred confidence intervals the normality tests
for all of these parameters were performed. Table 6 summarizes the statistical information
about the normality tests, average values and confidence intervals for each month. The values
of the A-squared parameter shows that the data is non-normal (D'Agostino,1986). According
to D'Agostino (1986), the cricital value of the  A-squared parameter, for  a 95% confidence
level, is 0.752. The values of this parameter presented in Table 6 are higher than this critical
value. So, there is a very strong evidence that the data is non-normal. This result is confirmed
by the p-values that are, in all cases, lower than 0.05 (significance level) implying the
rejection of the normality hypothesis. The Minitab software has an option to calculate the
confidence intervals for non-normal data. The results are presented in Table 6.

Table 6.Monthly statistical reports.

| | Variable | Simple size. N | Anderson-Darling Normality Test | | Mean | StDev | SE Mean | 95% CI |
|---|---|---|---|---|---|---|---|---|
| ⅃ | Hs[m] | 7687 | A-Squared: 40.63 | p-value <0.005 | 1.92191 | 0.50899 | 0.00581 | (1.91053; 1.93329) |

| | Parameter | N | A-Squared | p-value | Statistics |
|---|---|---|---|---|---|
| | **Tp[s]** | 7687 | A-Squared: 170.24 | p-value <0.005 | 10.7142  3.1631  0.0361 (10.6435; 10.7849) |
| | **P [kW/m]** | 7687 | A-Squared: 202.38 | p-value <0.005 | 23.513  13.783  0.157 ( 23.205; 23.821) |
| | **Hs [m]** | 7008 | A-Squared: 15.53 | p-value <0.005 | 1.87711 0.46451 0.00555 (1.86623; 1.88798) |
| **F** | **Tp[s]** | 7008 | A-Squared: 145.66 | p-value <0.005 | 10.4387  3.0192  0.0361 (10.3680;10.5094) |
| | **P [kW/m]** | 7008 | A-Squared: 208.03 | p-value <0.005 | 21.897  12.716  0.152 ( 21.599; 22.195) |
| | **Hs [m]** | 7689 | A-Squared: 21.63 | p-value <0.005 | 1.80126 0.43902 0.00501 (1.79144; 1.81107) |
| **M** | **Tp[s]** | 7689 | A-Squared: 70.03 | p-value <0.005 | 10.8515  2.8814  0.0329 (10.7871; 10.9159) |
| | **P [kW/m]** | 7689 | A-Squared: 131.79 | p-value <0.005 | 20.780  10.801  0.123 ( 20.538; 21.021) |
| | **Hs[m]** | 7440 | A-Squared: 36.30 | p-value <0.005 | 1.80543 0.38490 0.00446 (1.79668; 1.81417) |
| **A** | **Tp[s]** | 7440 | A-Squared: 118.55 | p-value <0.005 | 10.3233  2.7986  0.0324 (10.2597; 10.3869) |
| | **P[kW/m]** | 7440 | A-Squared: 161.64 | p-value <0.005 | 19.763   9.983   0.116 ( 19.536; 19.990) |
| | **Hs[m]** | 15376 | A-Squared: 29.32 | p-value <0.005 | 1.73386 0.31984 0.00258 (1.72881; 1.73892) |
| **M** | **Tp [s]** | 15376 | A-Squared: 491.92 | p-value <0.005 | 10.2287  3.0524  0.0246 (10.1804; 10.2769) |
| | **P[kW/m]** | 15376 | A-Squared: 258.45 | p-value <0.005 | 17.8068  7.9966  0.0645 (17.6804; 17.9332) |
| | **Hs[m]** | 14880 | A-Squared: 29.78 | p-value <0.005 | 1.64809 0.30307 0.00248 (1.64322; 1.65296) |
| **J** | **Tp [s]** | 14880 | A-Squared: 618.05 | p-value <0.005 | 10.1125  3.0069  0.0246 (10.0642; 10.1608) |
| | **P[kW/m]** | 14880 | A-Squared: 291.89 | p-value <0.005 | 16.0597  7.4576  0.0611 (15.9399; 16.1795) |
| **J** | **Hs[m]** | 15376 | A-Squared: 46.52 | p-value <0.005 | 1.59065 0.26830 0.00216 (1.58640; |

| | | | | | |
|---|---|---|---|---|---|
| | | | | | 1.59489) |
| | **Tp [s]** | 15376 | A-Squared: 849.41 | p-value <0.005 | 10.1592  2.8717  0.0232 (10.1138; 10.2046) |
| | **P[kW/m]** | 15376 | A-Squared: 254.08 | p-value <0.005 | 15.0375  6.6470  0.0536 (14.9324; 15.1425) |
| | **Hs[m]** | 7688 | A-Squared: 27.43 | p-value <0.005 | 1.57631  0.26316  0.00300 (1.57043; 1.58219) |
| **A** | **Tp[s]** | 7688 | A-Squared: 337.55 | p-value <0.005 | 10.2906  2.9649  0.0338 (10.2243; 10.3569) |
| | **P[kW/m]** | 7688 | A-Squared: 174.44 | p-value <0.005 | 15.1119  7.2471  0.0827 (14.9499; 15.2740) |
| | **Hs[m]** | 7440 | A-Squared: 13.53 | p-value <0.005 | 1.59887  0.27965  0.00324 (1.59251; 1.60522) |
| **S** | **Tp[s]** | 7440 | A-Squared: 204.76 | p-value <0.005 | 10.2960  2.8409  0.0329 (10.2315; 10.3606) |
| | **P[kW/m]** | 7440 | A-Squared: 143.42 | p-value <0.005 | 15.4316  7.0104  0.0813 (15.2723; 15.5910) |
| | **Hs[m]** | 7687 | A-Squared: 24.60 | p-value <0.005 | 1.60069  0.33400  0.00381 (1.59322; 1.60816) |
| **O** | **Tp[s]** | 7687 | A-Squared: 61.21 | p-value <0.005 | 10.8908  2.8969  0.0330 (10.8261; 10.9556) |
| | **P[kW/m]** | 7687 | A-Squared: 188.74 | p-value <0.005 | 16.5502  8.6290  0.0984 (16.3573; 16.7431) |
| | **Hs[m]** | 7440 | A-Squared: 45.13 | p-value <0.005 | 1.65678  0.39347  0.00456 (1.64784; 1.66573) |
| **N** | **Tp[s]** | 7440 | A-Squared: 47.37 | p-value <0.005 | 11.0808  2.9679  0.0344 (11.0133; 11.1482) |
| | **P[kW/m]** | 7440 | A-Squared: 212.76 | p-value <0.005 | 18.439  11.008  0.128 ( 18.189; 18.689) |
| | **Hs[m]** | 7688 | A-Squared: 65.15 | p-value <0.005 | 1.80871  0.45569  0.00520 (1.79852; 1.81890) |
| **D** | **Tp[s]** | 7688 | A-Squared: 110.55 | p-value <0.005 | 10.7810  3.1661  0.0361 (10.7102; 10.8518) |
| | **P[kW/m]** | 7688 | A-Squared: 298.54 | p-value <0.005 | 21.213  13.252  0.151 ( 20.917; 21.509) |

The energy from NCs is a time-varying quantity. Thus, to estimate this parameter it is necessary to achieve the minimum sampling dimension to guarantee its temporal representativeness. As the wave regime is the only parameter that causes the variation in the energy content produced by NCs, we assume that the minimum sampling size necessary to characterize the monthly average power on waves is equal to the minimum sampling size to characterize the monthly average power emanating from the NC. Further, this minimum sampling dimension is converted in numbers of days for monitoring the NCs in order to achieve the temporal representativeness of the power data. In this way, using the Minitab software, the minimum number of sample points, for average monthly power, was calculated admitting a 0.85 power factor, a significance level equal to 0.05 and a value of 3kW/m for margin of error. This margin of error was assumed taking into account the possibility to completing all measurements in one year. In this context, lower margin of error implies higher number of sample points. Table 7 show the standard deviations, the minimum sampling size to guarantee the representativeness of the values of the monthly average power and, consequently, the number of days to carry out the experiments on the Natural Caves in order to ensure the correct values of the average power extracted from these natural infrastructures.

It is important to note that during the spring (March to May) and summer (June to August) the minimum numbers of days of measurements are lowers in comparison with the rest of the months. The reason for this finding is associated with the nature of the wave data for the referred months. That is, during the spring and summer the wave data present low dispersion as it is possible to see through the values of the standard deviation in Table 7, indicating that the wave energy resources are most stable during these periods of the year. Therefore, the minimum sample size for characterizing these wave data is lower than the rest of the months for which the standard deviations are higher.

Table 7. Minimum sampling size and the corresponding numbers of days of measurements

| Power Factor: 0.85; Margin of Error: 3 kW/m;  Significance level: $\alpha = 0.05$ | | | |
|---|---|---|---|
| Months | Standard deviation, σ | Minimum sampling size, n | Numbers of days (for 3 h time step) |
| J | 13.25 | 178 | 23 |
| F | 11.01 | 123 | 16 |
| M | 8.63 | 77 | 10 |

| | | | |
|---|---|---|---|
| A | 7.01 | 51 | 7 |
| M | 7.25 | 55 | 7 |
| J | 6.65 | 47 | 6 |
| J | 7.46 | 58 | 8 |
| A | 7.99 | 66 | 9 |
| S | 9.98 | 102 | 13 |
| O | 10.80 | 119 | 15 |
| N | 12.78 | 165 | 21 |
| D | 13.78 | 192 | 24 |

**5   Conclusion**

The most common sea state in Cape-Verde occurs 20.81% of time, featuring peak periods from 6-9 s and significant wave height from 1.5-2 m. For period between 1979 and 2009,

78.03% of the waves present wave height between 1 and 2 m.

January and December are the most energetic months and July is the least energetic month.

The monthly wave power decreases from January to July and increases again to December.

Through the Coefficient of Variation of Power (COVP) it is possible to conclude that the wave resource is stable, with COVP between 0.46 and 0.66.

The MVI parameter shows that the wave resource can be considered relatively stable (MVI

<1.2) from monthly average power point of view.

The monthly average and the annual time-series are stationeries over time. The confidence intervals for all months were calculated using the Minitab software.

The time-series of inter-annual average wave power shows some attenuation over the years, due to the occurrence of effect of aggregation. However, using the smoothing moving average curve it is possible to verify that, from 1999, inter-annual average wave power oscillate around of 18 kW/m.

This trend is confirmed by the analysis of 10 years of the inter-annual average power future values, using an appropriate ARIMA model generated automatically by R software.

The outliers, present in time-series of annual average power were identified and their influence in the value of inter-annual average power was quantified. Removing outliers from the annual time-series of power caused a maximum relative attenuation in the values of the inter-annual average power between 1.85 and 13%.

The minimum recording time of physical parameters associated with the NC operation are determined, for each month, under the assumption that the minimum sampling size necessary to characterize the monthly average power on waves is equal to the minimum sampling size to characterize the monthly average power emanating from the NC. In this context and for the Cape-Verde Wave Regime, the minimum sampling size and the corresponding numbers of days of measurements are given in Table 7. During the spring and summer the wave resources are more stable than the rest of the year and, therefore, the minimum numbers of day for monitoring the NCs are lower, in comparison with the rest of period of time.

**Acknowledgements**

We are grateful to Jackson Augusto Léger Monteiro for his important contribution to the realization of this work.

---

## Editor Comment (EC1) · J. M. Huthnance (Editor) · 17 Apr 2016

Dear Authors

You will have seen that the reviews are very critical. I must reinforce the Copernicus letter that the revised article must address the referees' comments. If the manuscript is eventually published in Ocean Science, the referee comments will remain accessible to readers who will be able to see whether the comments are addressed. Before that, however, referees or I will make that judgement.

Yours sincerely

John Huthnance